

**Feasibility analysis of using inverse modeling for estimating field-scale evapotranspiration in**
**maize and soybean fields from soil water content monitoring networks**
Foad Foolad[1], Trenton E. Franz[2], Tiejun Wang[2, 3], Justin Gibson[2], Ayse Kilic[1, 2], Richard G. Allen[4],
Andrew Suyker[2]
[1]Civil Engineering Department, University of Nebraska-Lincoln, USA
[2]School of Natural Resources, University of Nebraska-Lincoln, USA
[3]Institute of Surface-Earth System Science, Tianjin University, P.R. China
[4]Kimberly Research and Extension Center, University of Idaho, USA
*Keywords*: Evapotranspiration; soil water content; inverse modeling; soil hydraulic parameters;
Cosmic ray neutron probe
Corresponding author T.E. Franz (tfranz2@unl.edu)



**Abstract**
In this study the feasibility of using inverse vadose zone modeling for estimating field scale actual
evapotranspiration ($ET_a$) was explored at a long-term agricultural monitoring site in eastern
Nebraska. Data from both point scale soil water content sensors ($SWC$) and the area-average
technique of cosmic-ray neutron probes were evaluated against independent $ET_a$ estimates from a
co-located eddy covariance tower. While this methodology has been successfully used for estimates
of groundwater recharge it was critical to assess the performance of other components of the water
balance such as $ET_a$. In light of the recent evaluation of Land Surface Model (LSM) performance
from the plumber experiment, independent estimates of hydrologic state variables and fluxes are
critically needed benchmarks. The results here indicate reasonable estimates of daily and annual $ET_a$
from the point sensors but with highly varied soil hydraulic function parameterizations due to local
soil texture variability. The results of multiple soil hydraulic parameterizations leading to equally
good $ET_a$ estimates is consistent with the hydrological principle of equifinality. While this study
focused on one particular site the framework can be easily applied to other $SWC$ monitoring
networks across the globe. The value added products of groundwater recharge and $ET_a$ flux from the
$SWC$ monitoring networks will provide additional and more robust benchmarks for the validation of
LSM that continue to improve their forecast skill. In addition, the value added products of
groundwater recharge and $ET_a$ often have more direct impacts on societal decision making than $SWC$
alone. Water flux impacts human decision making from policies on the long-term management of
groundwater resources (recharge), to yield forecasts ($ET_a$), and to optimal irrigation scheduling ($ET_a$).
Illustrating the societal benefits of $SWC$ monitoring is critical to insure the continued operation and
expansion of these public datasets.





## 1. Introduction

Evapotranspiration ($ET$) is an important component in terrestrial water and surface energy

balance. In the United States, $ET$ comprises about 75% of annual precipitation, while in arid and

semiarid regions $ET$ comprises more than 90% of annual precipitation (Zhang et al., 2001; Glenn et

al., 2007; Wang et al., 2009a). As such, an accurate estimation of $ET$ is critical in order to predict

changes in hydrological cycles and improve water resource management (Suyker et al., 2008;

Anayah and Kaluarachchi, 2014). Given the importance of $ET$, an array of measurement techniques

at different temporal and spatial scales have been developed (c.f., Maidment, 1992; Zhang et al.,

2014), including lysimeter, Bowen ratio, Eddy-Covariance (EC), and satellite-based surface energy

balance approaches. However, simple, low-cost, and accurate field-scale measurements of actual $ET$

($ET_a$) still remain a challenge due to the uncertainties of available estimation techniques (Wolf et al.,

2008; Li et al., 2009; Senay et al., 2011; Stoy, 2012). For instance, field techniques, such as EC and

Bowen ratio, can provide relatively accurate estimation of local $ET_a$, but are often cost prohibitive

for wide-spread use beyond research applications (Baldocchi et al., 2001; Irmak, 2010). By

comparison, satellite-based remote sensing techniques are far less costly for widespread spatial

coverage (Allen et al., 2007), but are limited by their accuracy, temporal sampling frequency (e.g.,

Landsat 8 has a 16-day overpass), and technical issues that further limit temporal sampling periods

(e.g., cloud coverage during overpass) (Chemin and Alexandridis, 2001; Xie et al., 2008; Li et al.,

2009; Kjaersgaard et al., 2012).

As a complement to the above mentioned techniques, recent studies have used process-based

vadose zone models (VZMs) for estimating field-scale $ET_a$ with reasonable success, particularly in

arid and semi-arid areas (Twarakavi et al., 2008; Izadifar and Elshorbagy, 2010; Galleguillos et al.,

2011; Wang et al., 2016). Although VZMs are time and cost effective for estimating field-scale $ET_a$,





they generally require complex model parameterizations and inputs, some of which are not readily
available (e.g., soil hydraulic parameters and plant physiological parameters; Wang et al., 2016). In
order to address the issue of missing soil hydraulic parameters, a common approach is to use
pedotransfer functions to convert readily available soil information (e.g., texture, bulk density, etc.)
to soil hydraulic parameters (Wösten et al., 2001); however, significant uncertainties are usually
associated with this method for estimating local scale water fluxes (Wang et al., 2015). In fact,
Nearing et al. (2016) identified soil hydraulic property estimation as the largest source of information
lost when evaluating different land surface modeling schemes versus a soil moisture benchmark.
Poor and uncertain parameterization of soil hydraulic properties are a clear weakness of land surface
models (LSMs) predictive skill in sensible and latent heat fluxes (Best et al., 2015). This problem
will continue to compound with the continuing spatial refinement of hyper-resolution LSMs grid
cells to less than 1 km (Wood et al., 2011).
In order to address the challenge of field scale estimation of soil hydraulic properties, here
we utilize inverse modeling for estimating soil hydraulic parameters based on field measurements
of soil water content (*SWC*) (c.f. Hopmans and Šimunek, 1999; Ritter et al., 2003). While VZM-
based inverse modeling approaches have already been examined for estimating groundwater
recharge (e.g., Jiménez-Martínez et al., 2009; Andreasen et al., 2013; Min et al., 2015; Ries et al.,
2015; Turkeltaub et al., 2015; Wang et al., 2016), its application for $ET_a$ estimation has not been
adequately tested. Moreover, we note that simultaneous estimation of *SWC* states and surface energy
fluxes within LSMs is complicated by boundary conditions, model parameterization, and model
structure (Nearing et al., 2016). With the incorporation of regional soil datasets in LSMs like Polaris
(Chaney et al., 2016), effective strategies for estimating ground truth soil hydraulic properties from



existing *SWC* monitoring networks (e.g., SCAN, CRN, COSMOS, State/National Mesonets, c.f. Xia
et al. (2015)) will become critical for continuing to improve the predictive skill of LSMs.

The aim of this study is to examine the feasibility of using inverse VZM modeling for

estimating field scale $ET_a$ based on long-term local meteorological and *SWC* observations for an
Ameriflux (Baldocchi et al., 2001) eddy covariance site in eastern Nebraska, USA. The remainder
of the paper is organized as follows. In the methods section we will describe the widely used VZM,
Hydrus-1D (Simnunek et al., 2013), used to obtain soil hydraulic parameters. We will assess the
feasibility of using both profiles of in-situ *SWC* probes as well as the area-average *SWC* technique
from Cosmic-Ray Neutron Probes (CRNP). In the results section we will compare the calibrated
VZM with independent $ET_a$ estimates provided by EC observations. Finally, we note that while this
study focused on one particular study site in eastern Nebraska, the methodology can be easily
adapted to a variety of *SWC* monitoring networks across the globe (Xia et al., 2015).

**2. Materials and Methodology**
**2.1 Study Site**

The study site is located in eastern Nebraska, USA at the University of Nebraska Agricultural

and Development Center near Mead. The field site (US-Ne3, Figure 1a, 41.1797 N°, 96.4397° W)
is part of the Ameriflux Network (Baldocchi et al., 2001) and has been operating continually since
2001. The regional climate is of a continental semiarid type with a mean annual precipitation of 784
mm/year (According to the Ameriflux US-Ne3 website). According to the Web Soil Survey Data
(Soil Survey Staff, 2016), the soils at the site are comprised mostly of silt loam and silty clay loam.
Soybean and maize are rotationally grown at the site under rainfed conditions, with the growing



season beginning in early May and ending in October (Kalfas et al., 2011). Since 2001, crop
management practices (i.e., planting density, cultivars, irrigation, and herbicide and pesticide
applications) have been applied in accordance with standard best management practices prescribed
for production-scale maize systems (Suyker et al., 2008). More detailed information about site
conditions can be found in Suyker et al. (2004) and Verma et al. (2005).

An EC tower was constructed at the center of the field (Figure 1 and Figure 2a), which

continuously measures water, energy, and $CO_2$ fluxes (e.g., Baldocchi et al., 1988). In this study,
hourly latent heat flux measurements were integrated to daily values and then used for calculating
daily $ET_a$ integrated over the field scale. Detailed information on the EC measurements and
calculation procedures for $ET_a$ are given in Suyker and Verma (2009). Hourly air temperature,
relative humidity, horizontal wind speed, net radiation, and precipitation were also measured at the
site. Destructive measurements of leaf area index ($LAI$) were made every 10 to 14 days during the
growing season at the study site (Suyker et al., 2005). We note that the $LAI$ data were linearly
interpolated to provide daily estimates. Theta probes (Delta-T Devices, Cambridge, UK) were
installed at 4 locations in the study field with measurement depths of 10, 25, 50, and 100 cm at each
location to monitor hourly $SWC$ in the root zone (Suyker et al., 2008). Here, we denote these four
locations as TP 1 (41.1775° N, 96.4442° W), TP 2 (41.1775° N, 96.4428° W), TP 3 (41.1775° N,
96.4402° W), and TP 4 (41.1821° N, 96.4419° W) (Figure 1b). Daily precipitation ($P$) and reference
evapotranspiration ($ET_r$) computed for the tall (alfalfa) reference crop using the ASCE standardized
Penman-Monteith equation (ASCE-EWRI 2005) are shown in Figure 3 for the study period (2007–
2012) at the study site.

In addition, a Cosmos-Ray Neutron Probe (CRNP, model CRS 2000/B, HydroInnova LLC,

Albuquerque, NM, USA) (41.1798 N°, 96.4412° W) was installed near the EC tower (Figure 1b and




2b) on 20 April 2011. The CRNP measures hourly moderated neutron counts (Zreda et al., 2008,
2012), which are converted into *SWC* following standard correction procedures and calibration
methods (c.f., Zreda et al., 2012). In addition, the changes in above-ground biomass were removed
from the CRNP estimates of *SWC* following Franz et al. (2015). The CNRP measurement depth
(Franz et al., 2012) at the site varies between 15-40 cm, depending on *SWC*. Note for simplicity in
this analysis we assume the CRNP has an effective depth of 20 cm (mean depth of 10 cm) for all
observational periods. For a more general integration of CRNP data into the NOAH LSM data
assimilation framework, we refer to the work of Shuttleworth et al. (2013) and Rosolem et al. (2014).
The areal footprint of the CRNP is ~250+/-50 m radius circle (see Desilets and Zreda (2013) and
Kohli et al. (2015) for details). Here we assume for simplicity the EC and CRNP footprints are both
representative of the areal-average field conditions.

**2.2. Model setup**
**2.2.1 Vadose Zone Model**

The Hydrus-1D model (Šimunek et al., 2013), which is based on the Richards equation, was

used to calculate $ET_a$. The setup of the Hydrus-1D model is explained in details by Jiménez-Martínez
et al. (2009), Min et al. (2015), and Wang et al. (2016), and only a brief description of the model
setup is provided here. Given the measurement depths of the Theta Probes, the simulated soil profile
length was chosen to be 175 cm with 176 nodes at 1 cm intervals. An atmospheric boundary
condition with surface runoff was selected as the upper boundary. This allowed the occurrence of
surface runoff when precipitation rates were higher than soil infiltration capacity or if the soil
became saturated. According to a nearby USGS monitoring well (Saunders County, NE, USGS





411005096281502, ~2.7 km away), the depth to water tables was greater than 12 m during the study
period. Therefore, free drainage was used as the lower boundary condition.

Daily $ET_r$ was calculated using the ASCE Penman-Monteith equation for the tall (0.5 m)

ASCE reference (ASCE-EWRI, 2005), and daily potential evapotranspiration ($ET_p$) was calculated
according to FAO 56 (Allen et al., 1998):
$$ET_p(t) = K_C(t) \times ET_r(t) \qquad (1)$$
where $Kc$ is a crop-specific coefficient at time $t$. The estimates of growth stage lengths and $Kc$ values
for maize and soybean suggested by Allen et al. (1998) and Min et al. (2015) were adopted in this
study. In order to partition daily $ET_p$ into potential transpiration $(T_p)$ and potential evaporation $(E_p)$
Beer's law was used as follows:
$$E_p(t) = ET_p(t) \times e^{-k \times LAI(t)} \qquad (2)$$
$$T_p(t) = ET_p(t) - E_p(t) \qquad (3)$$
where $k$ is an extiction coefficient with a value set to 0.5 (Wang et al., 2009b) and $LAI$ ($L^2/L^2$) is leaf
area index described above. The root water uptake, which was assumed to be equal to actual
transpiration, was simulated according to the Feddes model, based on $T_p$ and root density distribution
(Feddes et al., 1978). Since the study site has annual cultivation rotations between soybean and
maize, the root growth model from the Hybrid-Maize Model (Yang et al., 2004) was used to model
the root growth during the growing season:





$$\begin{cases} if\ D < MRD \\[2em] D = \dfrac{AGDD}{GDD_{Silking}} MRD \\[2em] else \\[2em] D = MRD \end{cases}$$
(4)

Where $D$ (cm) is plant root depth for each growing season day, $MRD$ is the maximum root depth
(assumed equal to 150 cm for maize and 120 cm for soybean in this study following Yang et al.
(2004)), $AGDD$ is the accumulated growing degree days, and $GDD_{Silking}$ is the accumulated $GDD$ at
the silking point (e.g., Accumulated plant $GDD$ approximatly 60-70 days after crop emergence).
$GDD$ for each growing season day was calculated as:
$GDD = \dfrac{T_{max} - T_{min}}{2} - T_{base}$                    (5)
where $T_{max}$ and $T_{min}$ are the maximum and minimum daily temperature (°C), respectively, and $T_{base}$
is the base temperature set to be 10° C following McMaster and Wilhelm (1997) and Yang et al.

(1997).


**2.2.2 Inverse modeling to estimate soil hydraulic parameters**

Inverse modeling was used to estimate soil hydraulic parameters for the van Genuchten-

Mualem model (Mualem, 1976; van Genuchten, 1980):
$\theta(h) = \begin{cases} \theta_r + \dfrac{\theta_s - \theta_r}{(1 + |\alpha h|^n)^m}, & h < 0 \\ \theta_s, & h \geq 0 \end{cases}$                    (6)



$K(S_e) = K_s \times S_e^{\ l} \times [1 - (1 - S_e^{\ 1/m})^m]^2$         (7)

where $\theta$ ($L^3/L^3$) is volumetric $SWC$; $\theta_r$ ($L^3/L^3$) and $\theta_s$ ($L^3/L^3$) are residual and saturated moisture
content, respectively; $h$ (L) is pressure head; $K$ (L/T) and $K_s$ (L/T) are unsaturated and saturated
hydraulic conductivity, respectively; and $S_e$ ($=(\theta-\theta_r)/(\theta_s-\theta_r)$) (-) is saturation degree. With respect to
the fitting factors, $\alpha$ (1/L) is inversely related to air entry pressure, $n$ (-) measures the pore size
distribution of a soil with $m=1-1/n$, and $l$ (-) is a parameter accounting for pore tortuosity and
connectivity.

Daily $SWC$ data from the four TP locations and CRNP location were used for the inverse

modeling. Based on the measurement depths of the TPs, the simulated soil columns were divided
into four layers (i.e., 0-15 cm, 15-35 cm, 35-75 cm, and 75-175 cm), which led to a total of 24
hydraulic parameters to be optimized ($\theta_r$, $\theta_s$, $\alpha$, $n$, $K_s$, and $L$). In order to efficiently optimize the
parameters, we used the method outlined in Turkeltaub et al. (2015). Specifically, the van Genuchten
parameters of the upper two layers were first optimized, while the parameters of the lower two layers
were fixed, since water contents of the lower layers changed more slowly and over a smaller range
than the upper layers. Then, the optimized van Genuchten parameters of the upper two layers were
kept fixed, while the parameters of the lower two layers were optimized. The process was continued
until there were no further improvements in the optimized hydraulic parameters or until the changes
in the lowest sum of squares were less than 0.1%. Given the sensitivity of the optimization results
to the initial guesses of soil hydraulic parameters in the Hydrus model, soil hydraulic parameters
from six soil textures were used as initial inputs for the optimizations at each location (Carsel and
Parish, 1988), including sandy clay loam, silty clay loam, loam, silt loam, silt, and clay loam. Based
on the length of available $SWC$ data from the TP measurements, the periods of 2007, 2008-2010,



and 2011-2012 were used as the spin-up, calibration, and validation periods, respectively. Moreover,
to minimize the impacts of freezing conditions on *SWC* measurements, data from January to March
of each calendar year were removed (based on available soil temperature data) from the
optimizations.

In addition to the TP profile observations, we used the CRNP area-average *SWC* in the

inverse procedure to develop an independent set of soil parameters. The CRNP was assumed to
provide *SWC* data with an average effective measurement depth of 20 cm at this study site. The
observation point was therefore set at 10 cm.  In addition, soil properties were assumed to be
homogeneous throughout the simulated soil column with a length of 175 cm. Since the CRNP was
installed in 2011 at the study site, the periods of 2011, 2012-2013, and 2014 were used as spin-up,
calibration, and validation periods, respectively, for the optimization procedure.

The lower and upper bounds on the van Genuchten parameters used are given in Table 1.

With respect to the goodness-of-fit assessment, four performance criteria were selected to evaluate
the model results, including R-squared ($R^2$), Mean Average Error (MAE), Root Mean Square Error
(RMSE), and the Nash-Sutcliffe Efficiency (NSE):
$$MAE = \frac{1}{n}\sum_{i=1}^{n}|P_i - O_i| \qquad (8)$$
$$RMSE = \sqrt{\frac{1}{n}\sum_{i=1}^{n}(P_i - O_i)^2} \qquad (9)$$
$$NSE = 1 - \frac{\sum_{i=1}^{n}(P_i - O_i)^2}{\sum_{i=1}^{n}(O_i - \bar{O}_i)^2} \qquad (10)$$
where *n* is the total number of *SWC* data points, $O_i$, and $P_i$, are respectively the observed and
simulated daily *SWC* on day *i*, and $\bar{O}_i$ is the observed mean value. Finally, based on the best scores



(i.e., lowest MAE and RMSE, and highest NSE scores), the best optimized soil parameter values at
each location were selected. Using the selected parameters, the Hydrus model was run in a forward
mode in order to estimate $ET_a$ between 2007 and 2012. We note 2004-2006 was used as a spin-up
period for the forward model.

**3. Results and Discussions**
**3.1 Inverse Vadose Zone Modeling Results**

The time series of the average *SWC* from the four TP locations along with one standard

deviation at each depth is plotted in Figure 4. Despite the small spatial scale in this study (~65 ha),
Figure 4 clearly shows that *SWC* varies considerably across the site, particularly during the growing
season. The comparison between *SWC* data from the CRNP and spatial average of *SWC* data at four
TP locations in the study field (*i.e.* average of 10 and 25 cm depths at each location) is presented in
Figure 5 and 6. The daily RMSE between the spatial average of the TPs and CRNP data is 0.037
cm$^3$/cm$^3$, which is consistent with other studies that reported similar values in semiarid shrublands
(Franz et al., 2012), German Forests (Bogena et al., 2013, Baatz et al., 2014), montane forests in
Utah (Lv et al., 2014), sites across Australia (Hawdon et al., 2014), and a mixed land use agricultural
site in Austria (Franz et al. 2016). Note we would expect lower RMSE (~<0.02 cm$^3$/cm$^3$) with
additional point sensors located at shallower depths and in more locations spatially. Never-the-less,
the consistent behavior between the spatial mean *SWC* of TPs and the CRNP allows us to explore
spatial variability of soil hydraulic properties within footprint using inverse modeling. This will be
described in the next sections. The study period (2007-2012, Figure 7) contained significant
interannual variability in precipitation. During the spin-up period in 2007, the annual precipitation





(942 mm) was higher than the mean annual precipitation (784 mm), 2008 was a wet year (997 mm),
2009-2011 were near average years (715 mm), and 2012 was a record dry year (427 mm) with a
widespread drought across the region. Therefore, both wet and dry years were considered in the
inverse modeling simulation period.

As an illustration, Figure 8 shows the daily observed and simulated *SWC* during the

calibration (2008–2010) and validation (2011–2012) periods at the TP 1 location (the simulation
results of the other three sites can be found in the supplemental Figures S1, S2, and S3). The results
of four performance criteria (e.g., $R^2$, MAE, RMSE, and NSE) between simulated and observed
*SWC* data at TPs and CRNP locations are presented in Tables 2 and 3.

Similar to previous studies (e.g., Jiménez-Martínez et al., 2009; Andreasen et al., 2013; Min

et al., 2015; Wang et al., 2016), the results of all the performance criteria at TPs locations show the
capability of inverse modeling in estimation of soil hydraulic parameters. The results of the
calibration period (2008-2010) indicate that the simulated and observed *SWC* values are in good
agreement throughout the whole period. However, the match between the simulated and observed
*SWC* are better in the shallower depths of 10 and 25 cm (Figure 8 and Table 2). In addition, the
simulated and observed data were well matched during the validation period (2011-2012), except
during the second half of 2012 when the extreme drought occurred. Reasons for this disagreement
in the observed and simulated *SWC* data will be discussed in the following sections.

The results of inverse modeling using the CRNP data indicate the feasibility of using these

data to estimate effective soil hydraulic parameters (Table 3). Based on the performance criteria
(Table 3), the simulated data are fairly well-matched with the observed *SWC* data during both the
calibration and validation periods. However, as the crops extracted water from deeper soil layers


and due to the fact that the CRNP observational depth is limited to near surface layers (~20 cm), it
is clear from the data that the comparison between the simulated and observed values deteriorates
over the growing season (Figure 9). The results suggest that it might be more appropriate to use the
CRNP data for inverse modeling during periods that are dominated by soil evaporation (Jana et al.,
2016) and/or for sites with shallow rooted vegetation only. Additional information from deeper soil
probes or more complex modeling approaches such as data assimilation techniques (Rosolem et al.,
2014, Renzullo et al., 2014) may be needed to fully utilize the CRNP data for the entire growing
season; however, this is beyond the scope of this study and future investigations are still needed.

Table 4 summarizes the optimized van Genuchten parameters for the four different depths

of the four TP locations and the single layer for the CRNP location. The optimized parameters were
then used to estimate $ET_a$ for the entire study period as an independent comparison to the EC $ET_a$
data. The results of the $ET_a$ evaluation will be discussed in the next section. According to the
simulation results (Table 4), in most of the soil layers, the TP 4 location possesses lower $n$, $K_s$, and
higher $\theta_r$ values than the other 3 locations (TPs 1-3), suggesting either underlying soil texture
variability in the field or texture dependent sensor sensitivity/calibration. As a validation for the
simulation results, the publicly available Web Soil Survey Data were used to explore whether the
optimized van Genuchten parameters from the inverse modeling (Figure 10 and Table 5) agreed with
the survey data. Based on the Web Soil Survey Data, the soil at the TP 4 location contains higher
clay percentage than the other locations. Meanwhile, the optimized parameters reflect the spatial
pattern of soil texture in the field as shown by the Web Soil Survey data (e.g., lower $n$ and $K_s$ values
and higher $\theta_r$ values at the TP 4 location with finer soil texture). Physically, finer-textured soils
generally have lower $K_s$ and higher $\theta_r$ values (Carsel and Parrish, 1988). Moreover, the shape factor
$n$ is indicative of pore size distributions of soils. In general, finer soils with smaller pore sizes tend





to have lower $n$ values (Carsel and Parrish, 1988). The observed *SWC* at the TP 4 location is
consistently higher than the average *SWC* of the other three locations (Figure S4 in supplemental
materials), which can be partly attributed to the higher $\theta_r$ values at the TP 4 location (Wang and
Franz, 2015). Overall, the obtained van Genuchten parameters from the inverse modeling are in
good agreement with the spatial distribution of soil texture in the study field, indicating the capability
of using inverse VZM modeling to infer soil hydraulic properties.

**3.2 Comparison of modeled $ET_a$ with observed $ET_a$**

Using the best fit soil hydraulic parameters for the four TP sites and the CRNP site, the

Hydrus-1D model was run in a forward mode to calculate $ET_a$ over the entire study period. The
simulated daily $ET_a$ was then compared with the independent EC $ET_a$ measurements using the same
four performance criteria that were used to evaluate the simulated *SWC* time series (Table 6). The
performance criteria results indicate that the simulated daily $ET_a$ is in a better agreement with EC
$ET_a$ measurements at the TP 1-3 locations than at the TP 4 and CRNP locations (Table 6 and Figure
11). However, based on the performance criteria from inverse modeling results and based on the
Web Soil Survey Data, one can conclude that partly due to the spatial heterogeneity of soil texture
in the study field, spatial variation in $ET_a$ rates likely exist over the field (e.g., less $ET_a$ occurs at the
TP 4 location than from the other part of the field). In addition, higher surface runoff can be expected
at the TP 4 location due to finer-textured soils and therefore less stored water to support $ET_a$.
According to the simulation results the average surface runoff at the TP 4 location was about 44.8
mm/year from 2007 to 2012, while the average surface runoff at the other three locations (TPs 1-3)
was around 10.6 mm/year.





Given that CRNPs have a limited observational depth and only one single soil layer was
optimized in the inverse model for the CRNP, one could expect the simulated daily $ET_a$ from the
CRNP to have larger uncertainty (e.g., RMSE of 1.26 mm/day at the CRNP location versus mean
RMSE value of 1.07 mm/day at TP locations). However, when the optimized soil parameters
obtained from the CRNP data were used to estimate $ET_a$, the model did simulate daily $ET_a$ fairly
well during non-growing and early growing seasons in comparison to the EC $ET_a$ measurements;
however, with the development of deeper root systems after mid-June, the simulation results of daily
$ET_a$ did slightly deteriorate (Figure 13).
On the annual scale, $ET_a$ measured by the EC tower accounted for 87% of annual $P$ recorded
at the site during the study period (Figure 7). Overall, the simulated annual $ET_a$ at all the TP and
CRNP locations is comparable to the annual $ET_a$ measured by the EC tower, except during 2012
(Table 7, Figure 12 and Figure 13), in which a severe drought occurred in the region. One
explanation is that the plants extract more water from deeper layers under extreme drought
conditions than what we defined as a maximum rooting depth (150 cm for maize and 120 cm for
soybean) for the model, thus limiting the VZM model's ability to estimate $ET_a$ accurately during the
drought year (2012). Given the fact that EC $ET_a$ estimation can have up to 20% uncertainty
(Massman and Lee, 2002, and Hollineger and Richardson, 2005), and accounting for the natural
spatial variability of $ET_a$ due to soil texture, the various $ET_a$ estimation techniques performed well.
In fact, it is difficult to identify which is the clear solution if any. These results are consistent with
the concept of equifinality in hydrologic modeling given the complexity of natural systems (Beven
and Freer, 2001). Moreover, the findings here are consistent with Nearing et al. (2016) that show
information lost in model parameters greatly affects the soil moisture comparisons against a
benchmark. However, soil parameterization was less important in the loss of information for the



comparisons of *ET*/latent energy against a benchmark. Fully resolving these issues remains a key
challenge to the land surface modeling community and the model's ability to make accurate
predictions (Best 2015).

## 346   4. Conclusions

In this study the feasibility of using inverse vadose zone modeling for field scale $ET_a$

estimation was explored at an agricultural site in eastern Nebraska. Both point *SWC* sensors (TP)
and area-average techniques (CRNP) were explored. This methodology has been successfully used
for estimates of groundwater recharge but it was critical to assess the performance of other
components of the water balance such as $ET_a$. The results indicate reasonable estimates of daily and
annual $ET_a$ but with varied soil hydraulic function parameterizations. The varied soil hydraulic
parameters were expected given the heterogeneity of soil texture at the site and consistent with the
principle of equifinality in hydrologic systems. We note that while this study focused on one
particular site, the framework can be easily applied to other networks of *SWC* monitoring across the
globe (Xia et al., 2015). The value added products of groundwater recharge and $ET_a$ flux from the
*SWC* monitoring networks will provide additional and more robust benchmarks for the validation of
LSM that continue to improve their forecast skill.

## 360   Acknowledgments

This research is supported financially by the Daugherty Water for Food Global Institute at

the University of Nebraska, NSF EPSCoR FIRST Award, the Cold Regions Research Engineering
Laboratory through the Great Plains CESU, and an USGS104b grant. We sincerely appreciate the



support and the use of facilities and equipment provided by the Center for Advanced Land
Management Information Technologies, School of Natural Resources and data from Carbon
Sequestration Program, the University of Nebraska-Lincoln. TEF would like to thank Eric Wood for
his inspiring research and teaching career. No doubt the skills TEF learned while at Princeton in
formal course work, seminars, and discussions with Eric will serve him well in his own career.



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

invasively at intermediate spatial scale using cosmic-ray neutrons. Geophysical Research
Letters, 35(21).





**List of Figures**

Figure 1. Study site (Mead Rainfed/US-Ne3) location in Nebraska (a) and locations of Eddy-Covariance Tower (EC), Cosmic-Ray Neutron Probe (CRNP) and Theta Probes (TPs) at the study site, 2014 (b).

Figure 2. Eddy-Covariance Tower (a) and Cosmic-Ray Neutron Probe (b) Located at the Mead Rainfed (US-Ne3) Site.

Figure 3. Daily precipitation ($P$) and reference evapotranspiration ($ET_r$) during the calibration (2008–2010) and validation (2011–2012) periods at the Mead Rainfed (US-Ne3) Site.

Figure 4. Annual precipitation ($P$) and annual actual evapotranspiration ($ET_a$) at the Mead Rainfed (US-Ne3) Site.

Figure 5. Time series of daily CRNP and spatial average TP $SWC$ ($\theta$) data.

Figure 6. Comparison between daily CRNP and spatial average TP $SWC$ ($\theta$) data.

Figure 7. Temporal evolution of daily $SWC$ ($\theta$) at different soil depths. The black lines represent daily mean $SWC$ ($\theta$) calculated from TPs in 4 different locations at study site and the blue areas indicate one standard deviation.

Figure 8. Daily observed and simulated $SWC$ ($\theta$) during the calibration (2008–2010) and validation (2011–2012) periods at TP 1 location.

Figure 9. Daily observed and simulated $SWC$ ($\theta$) during the calibration (2012–2013) and validation (2014) periods at the location of Cosmic-Ray Neutron probe.

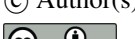



Figure 10. Variability of soil texture in the study field based on Web Soil Survey data (available at

http://websoilsurvey.sc.egov.usda.gov/App/HomePage.htm).

Figure 11. Annual actual Evapotranspiration ($ET_a$) estimation in different location at the study site

(2007-2012).

Figure 12. Mean Annual Actual Evapotranspiration ($ET_a$) estimation in different location at the

study site (2007-2012).

Figure 13. Cumulative simulated actual $ET$ versus cumulative observed actual $ET$ in different

locations at the study site (2007-2012).



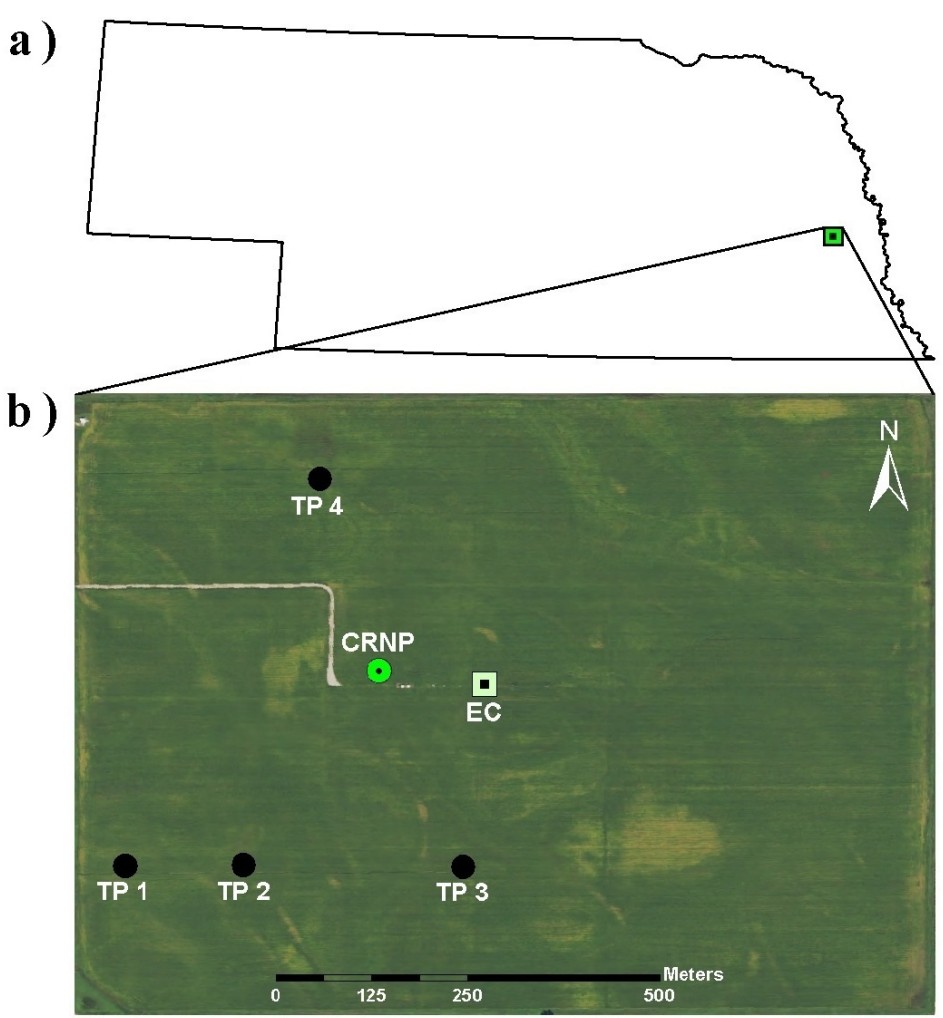


Figure 1. Study site (Mead Rainfed/US-Ne3) location in Nebraska (a) and locations of Eddy-
Covariance Tower (EC), Cosmic-Ray Neutron Probe (CRNP) and Theta Probes (TPs) at the
study site, 2014 (b).







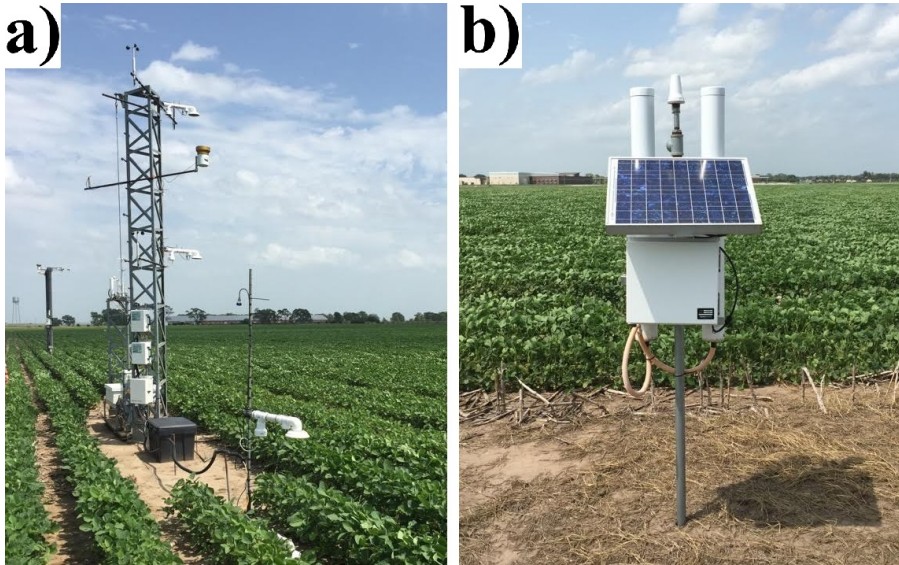


Figure 2. Eddy-Covariance Tower (a) and Cosmic-Ray Neutron Probe (b) Located at the Mead
Rainfed (US-Ne3) Site.



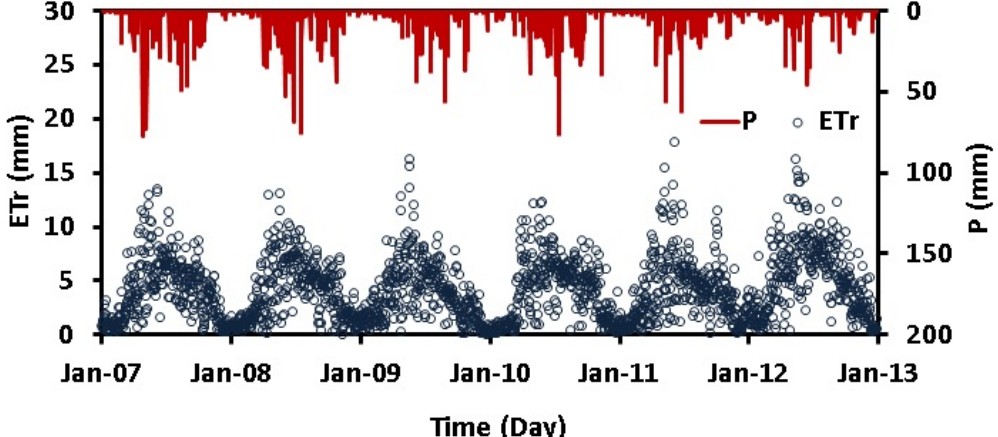

Figure 3. Daily precipitation (*P*) and reference evapotranspiration (*ET$_r$*) during the calibration
(2008–2010) and validation (2011–2012) periods at the Mead Rainfed (US-Ne3) Site.





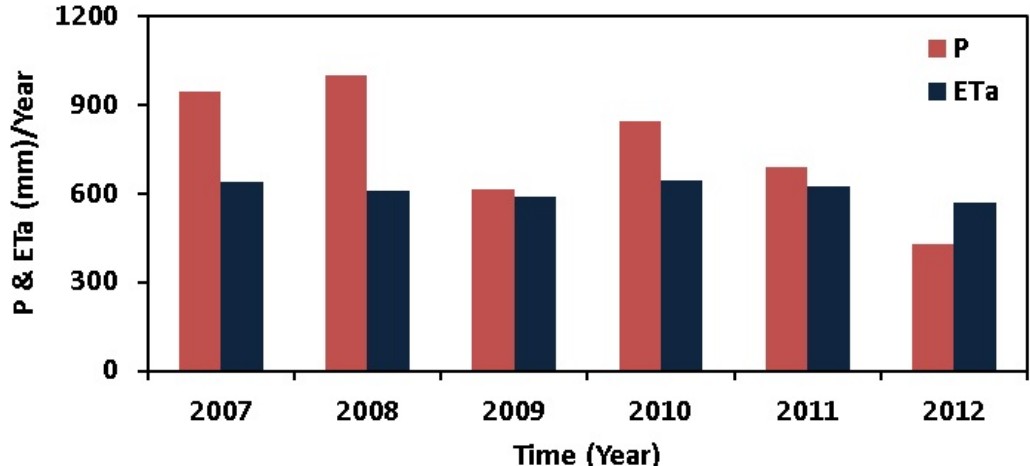


Figure 4. Annual precipitation ($P$) and annual actual evapotranspiration ($ET_a$) at the Mead Rainfed
(US-Ne3) Site.














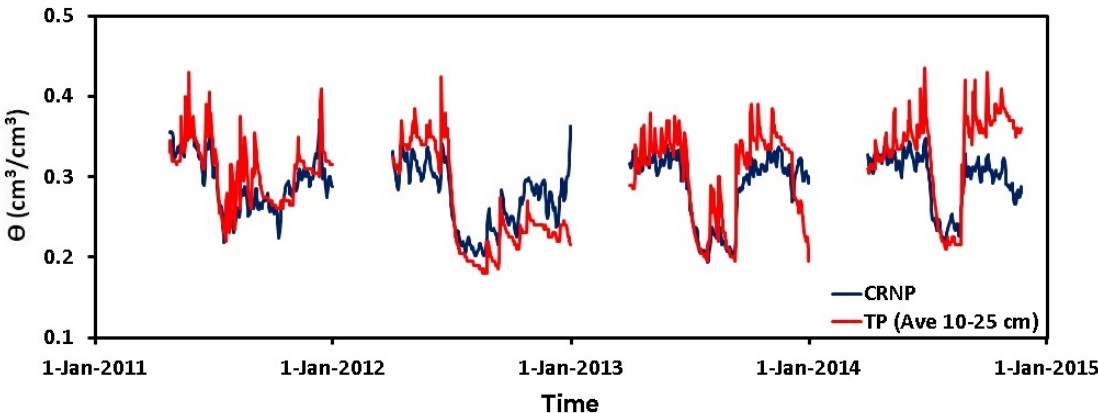

Figure 5. Time series of daily CRNP and spatial average TP *SWC* ($\theta$) data.












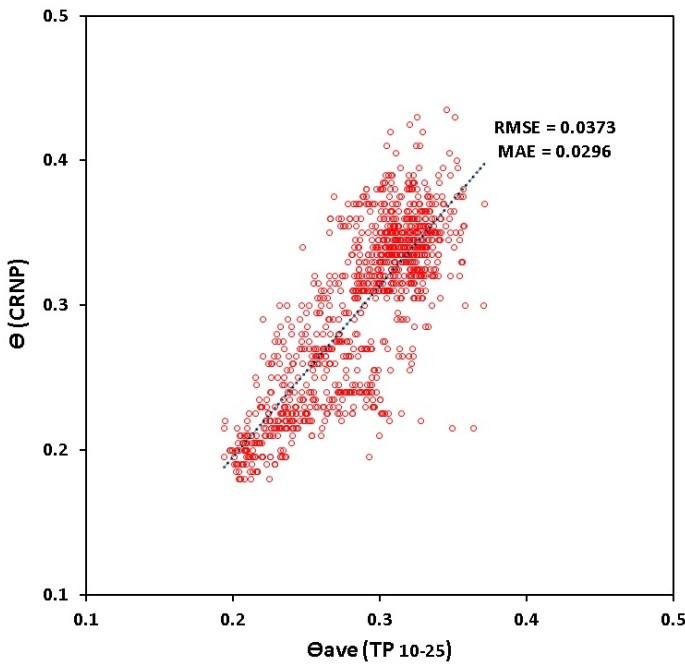


Figure 6. Comparison between daily CRNP and spatial average TP *SWC* (*θ*) data.



Figure 7. Temporal evolution of daily *SWC* (*θ*) at different soil depths. The black lines represent
daily mean *SWC* (*θ*) calculated from TPs in 4 different locations at study site and the blue areas
indicate one standard deviation.









Figure 8. Daily observed and simulated *SWC* (*θ*) during the calibration (2008–2010) and validation (2011–2012) periods at TP 1 location.





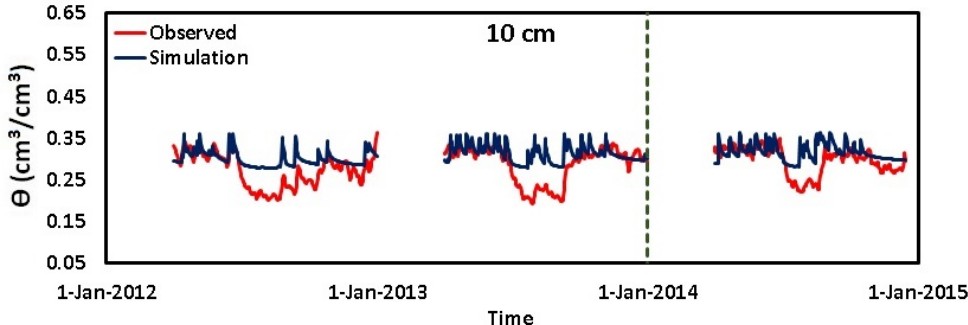


Figure 9. Daily observed and simulated *SWC* (*θ*) during the calibration (2012–2013) and validation
(2014) periods at the location of Cosmic-Ray Neutron probe.










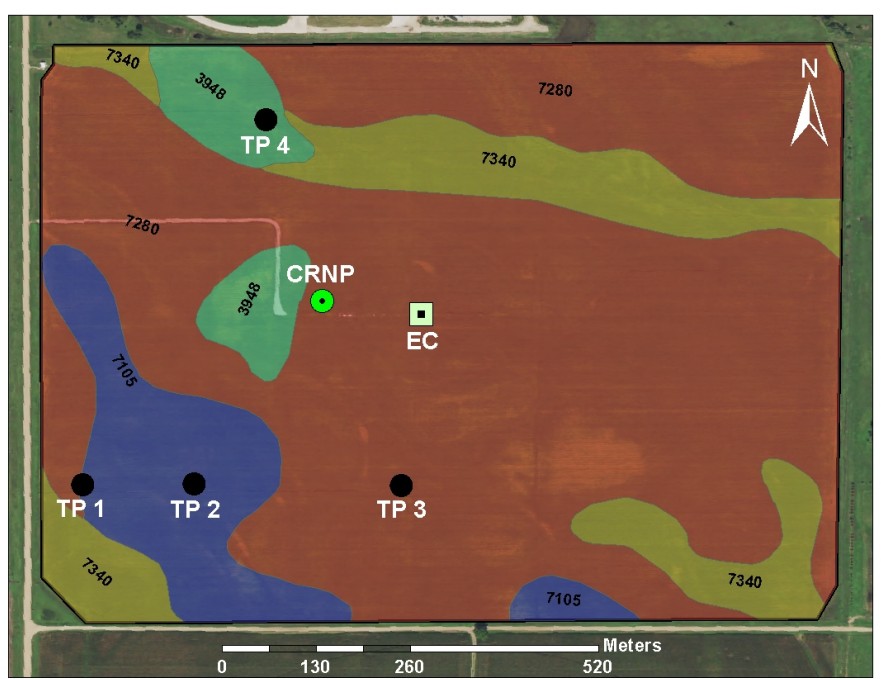


Figure 10. Variability of soil texture in the study field based on Web Soil Survey data (available at
http://websoilsurvey.sc.egov.usda.gov/App/HomePage.htm).





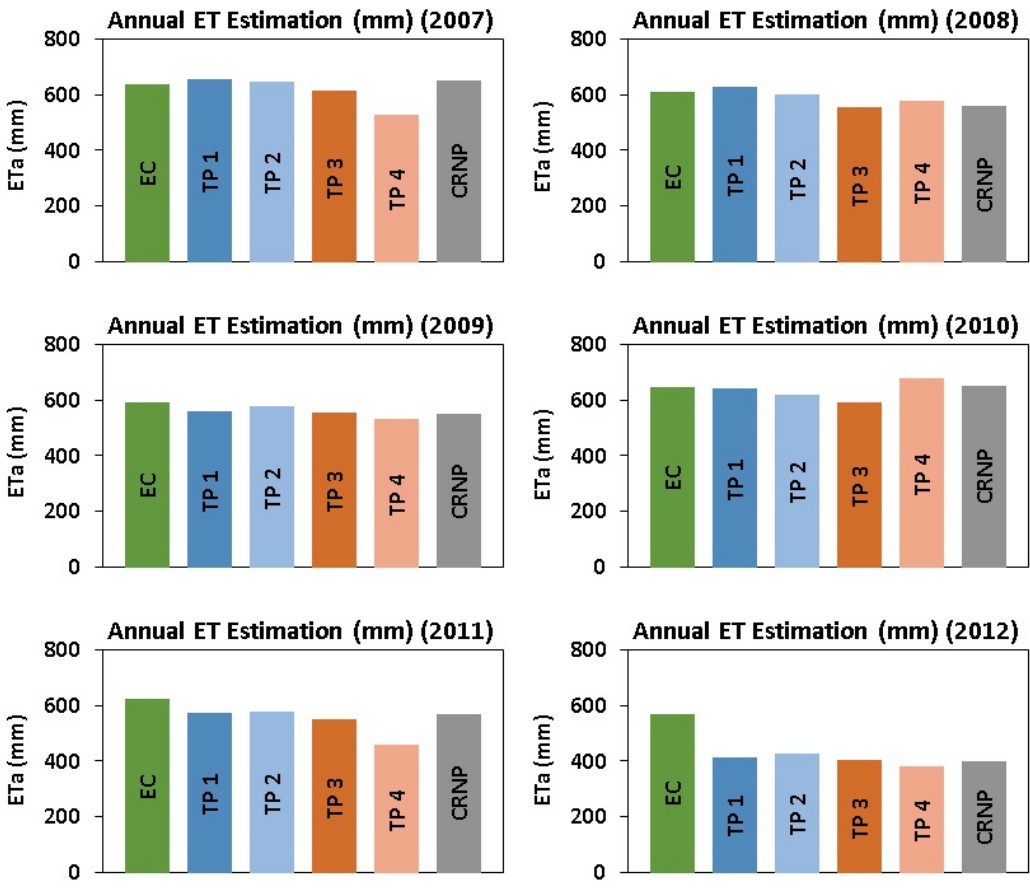


Figure 11. Annual actual Evapotranspiration ($ET_a$) estimation in different location at the study site
(2007-2012).









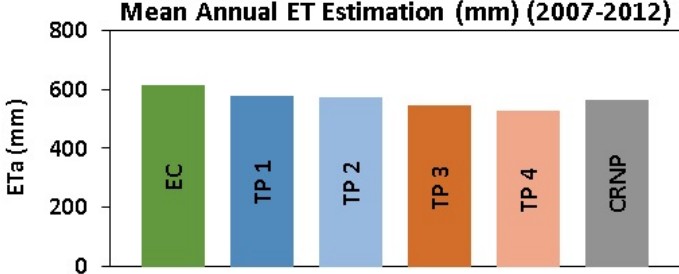


Figure 12. Mean Annual Actual Evapotranspiration ($ET_a$) estimation in different location at the
study site (2007-2012).





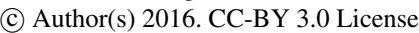


Figure 13. Cumulative simulated actual *ET* versus cumulative observed actual *ET* in different
locations at the study site (2007-2012).




**List of Tables**





Table 1. Bounds of the van Genuchten parameters used for inverse modeling.

| Soil Parameter | $\theta_r$ (-) | $\theta_s$ (-) | $\alpha$ (1/cm) | $n$ (-) | $K_s$ (cm/day) | $L$ (-) |
|:---:|:---:|:---:|:---:|:---:|:---:|:---:|
| Range | 0.03–0.30 | 0.3–0.6 | 0.001–0.200 | 1.01–6.00 | 1–200 | -1–1 |




















Table 2. Goodness-of-fit measures for simulated and observed *SWC* data at different depths during
the calibration period (2008 to 2010) and validation period (2011-2012) at TPs locations.

| Location | Depth (cm) | Calibration Period (2008-2010) | | | | Validation Period (2011-2012) | | | |
|---|---|---|---|---|---|---|---|---|---|
| | | $R^2$ | MAE | RMSE | NSE | $R^2$ | MAE | RMSE | NSE |
| TP 1 | 10 | 0.542 | 0.024 | 0.036 | 0.533 | 0.532 | 0.016 | 0.033 | 0.503 |
| | 25 | 0.742 | 0.014 | 0.022 | 0.739 | 0.716 | 0.029 | 0.040 | 0.486 |
| | 50 | 0.409 | 0.013 | 0.023 | 0.407 | 0.603 | 0.041 | 0.074 | 0.157 |
| | 100 | 0.352 | 0.015 | 0.022 | 0.343 | 0.419 | 0.027 | 0.038 | 0.358 |
| TP 2 | 10 | 0.330 | 0.044 | 0.066 | 0.305 | 0.287 | 0.047 | 0.061 | 0.052 |
| | 25 | 0.623 | 0.010 | 0.020 | 0.604 | 0.718 | 0.038 | 0.055 | 0.135 |
| | 50 | 0.551 | 0.015 | 0.026 | 0.074 | 0.683 | 0.040 | 0.055 | 0.202 |
| | 100 | 0.424 | 0.019 | 0.027 | -2.055 | 0.344 | 0.048 | 0.073 | -0.473 |
| TP 3 | 10 | 0.269 | 0.034 | 0.051 | 0.256 | 0.534 | 0.086 | 0.102 | -4.265 |
| | 25 | 0.512 | 0.011 | 0.017 | 0.509 | 0.852 | 0.010 | 0.015 | 0.793 |
| | 50 | 0.549 | 0.015 | 0.023 | -0.214 | 0.658 | 0.022 | 0.033 | 0.652 |
| | 100 | 0.238 | 0.018 | 0.029 | -3.156 | 0.669 | 0.018 | 0.025 | 0.178 |
| TP 4 | 10 | 0.412 | 0.029 | 0.044 | 0.406 | 0.580 | 0.051 | 0.071 | -0.116 |
| | 25 | 0.434 | 0.016 | 0.025 | 0.350 | 0.594 | 0.029 | 0.042 | 0.490 |
| | 50 | 0.151 | 0.009 | 0.015 | -13.400 | 0.443 | 0.041 | 0.073 | 0.036 |
| | 100 | 0.001 | 0.013 | 0.021 | -12.058 | 0.292 | 0.026 | 0.039 | 0.238 |









Table 3. Goodness-of-fit measures for simulated and observed *SWC* data during the calibration
period (2012 to 2013) and validation period (2014) at CRNP location.

| Location | Depth (cm) | Calibration Period (2012-2013) | | | | Validation Period (2014) | | | |
|---|---|---|---|---|---|---|---|---|---|
| | | $R^2$ | MAE | RMSE | NSE | $R^2$ | MAE | RMSE | NSE |
| CRNP | 10 | 0.352 | 0.024 | 0.038 | -0.059 | 0.083 | 0.021 | 0.034 | -0.454 |

















Table 4. Optimized van Genuchten parameters in different locations at the study site.

| Location | Depth (cm) | $\theta_r$ (-) | $\theta_s$ (-) | $\alpha$ (1/cm) | $n$ (-) | $K_s$(cm/day) | $L$ (-) |
|---|---|---|---|---|---|---|---|
| TP 1 | 0-15 | 0.134 | 0.423 | 0.027 | 1.475 | 8.119 | 0.546 |
| | 15-35 | 0.136 | 0.408 | 0.007 | 1.345 | 11.540 | 0.480 |
| | 35-75 | 0.191 | 0.448 | 0.024 | 1.097 | 8.057 | 0.285 |
| | 75-175 | 0.071 | 0.430 | 0.025 | 1.069 | 9.807 | 0.364 |
| TP 2 | 0-15 | 0.211 | 0.446 | 0.027 | 1.567 | 8.120 | 1.000 |
| | 15-35 | 0.197 | 0.434 | 0.006 | 1.191 | 8.655 | 0.022 |
| | 35-75 | 0.110 | 0.424 | 0.015 | 1.239 | 4.605 | 0.723 |
| | 75-175 | 0.109 | 0.408 | 0.020 | 1.302 | 6.780 | 0.000 |
| TP 3 | 0-15 | 0.281 | 0.464 | 0.035 | 1.487 | 7.096 | 0.400 |
| | 15-35 | 0.072 | 0.402 | 0.012 | 1.085 | 29.960 | 0.353 |
| | 35-75 | 0.081 | 0.498 | 0.037 | 1.128 | 24.440 | 0.527 |
| | 75-175 | 0.085 | 0.500 | 0.039 | 1.147 | 17.540 | 0.496 |
| TP 4 | 0-15 | 0.082 | 0.481 | 0.034 | 1.172 | 7.773 | 0.953 |
| | 15-35 | 0.200 | 0.426 | 0.013 | 1.217 | 14.060 | 0.044 |
| | 35-75 | 0.250 | 0.477 | 0.009 | 1.079 | 1.045 | 0.353 |
| | 75-175 | 0.200 | 0.487 | 0.012 | 1.070 | 1.454 | 0.985 |
| CRNP | 0-15 | 0.102 | 0.369 | 0.019 | 1.075 | 6.450 | 0.555 |






Table 5. Variability of soil texture in the study field based on Web Soil Survey data
(http://websoilsurvey.sc.egov.usda.gov/App/HomePage.htm).

| Map Unit Symbol | Map Unit Name | Clay (%) | Silt (%) | Sand (%) | Hectors in Field | Percent of Field |
|---|---|---|---|---|---|---|
| 3948 | Fillmore silt loam, terrace, occasionally ponded | 41.7 | 51.0 | 7.3 | 3.24 | 4.9% |
| 7105 | Yutan silty clay loam, terrace, 2 to 6 percent slopes, eroded | 25.8 | 59.4 | 14.8 | 6.88 | 10.3% |
| 7280 | Tomek silt loam, 0 to 2 percent slopes | 32.3 | 61.6 | 6.1 | 47.23 | 70.8% |
| 7340 | Filbert silt loam, 0 to 1 percent slopes | 41.4 | 51.7 | 6.9 | 9.34 | 14.0% |
| Total Area of Field | | | | | 66.69 | 100.0% |















Table 6. Goodness-of-fit measures for simulated and observed daily $ET_a$ during the simulation
period (2007-2012) at study site.

| Location | $R^2$ | MAE | RMSE | NSE |
|---|---|---|---|---|
| TP 1 | 0.652 | 0.696 | 1.062 | 0.618 |
| TP 2 | 0.754 | 0.610 | 0.907 | 0.746 |
| TP 3 | 0.751 | 0.601 | 0.904 | 0.728 |
| TP 4 | 0.413 | 0.878 | 1.387 | 0.168 |
| CRNP | 0.499 | 0.787 | 1.259 | 0.349 |

















Table 7. Summary of simulated yearly and average actual evapotranspiration ($ET_a$) (mm) and
observed yearly and average actual evapotranspiration ($ET_a$) (mm) from Eddy-Covariance
tower during 2007 to 2012.

| Location | Year | | | | | | |
|---|---|---|---|---|---|---|---|
| | 2007 | 2008 | 2009 | 2010 | 2011 | 2012 | Average |
| EC | 656.8 | 608.4 | 589.7 | 646.1 | 622.2 | 570.1 | 612.5 |
| TP 1 | 646.1 | 629.0 | 559.8 | 642.1 | 573.9 | 415.5 | 579.5 |
| TP 2 | 614.3 | 598.4 | 576.7 | 620.5 | 576.9 | 429.5 | 574.7 |
| TP 3 | 529.0 | 556.1 | 556.4 | 590.4 | 549.8 | 405.2 | 545.4 |
| TP 4 | 652.2 | 576.1 | 529.9 | 677.3 | 458.2 | 381.2 | 525.3 |
| CRNP | 656.8 | 559.6 | 549.9 | 652.8 | 570.7 | 400.1 | 564.2 |
