# Peer review of "Feasibility analysis of using inverse modeling for estimating field-scale evapotranspiration in maize and soybean fields from soil water content monitoring networks"

_Hydrology and Earth System Sciences, 2016_

## Referee Comment (RC1) · Anonymous Referee #1 · 9 Sep 2016

hess-2016-437

The study addresses an important issue of enlarging the data sets available for LSM validation by estimating AET from SWC measurements. Also the underlying idea that recharge and AET data are generally more valuable to society than SWC alone justifies this field of research. The manuscript is very well written.

However the inverse methodology description is very weak. There is no description of which search method is used! What is the combined objective function? A detailed sensitivity analysis has to be given, especially in light of the mentioned problems

of equifinality. It is extremely unlikely that all 24 parameters are sensitive and justify optimization. Also inverse modelling offers the opportunity to provide the reader with an estimate of the confidence intervals for each estimated parameter, which will also reveal the sensitivity and associated uncertainty.

The results on simulated SWC seems to be reasonable from a SWC perspective, but its important to also address the certainty/robustness and likelihood of the estimated soil parameters. Are they random parameter picks from an equifinal problem or are they physically reasonable and do their mutual differences fit into field/lab measurements (I assume soil samples exists from the sites) ? The author have attempted to validate the spatial distribution of the estimated soil parameters based on a soil map, which is highly appreciated. However it would have been interesting to utilize this information for regionalizing the soil parameters and thereby limiting the number of free parameters in the calibration. Likewise the soilmap could have been used to upscale the AET simulations to the field scale by including the soilmap instead of a simple average of the four points.

The results of the AET simulations seem to be very poor. I miss a critical view on the results regarding lacking ability to simulate even interannual variability (fig 11) and perhaps more importantly the apparently complete lack of predictive capability on the daily scale. The performance metrics in Table 6 indicate good R2 and NSE, but that correlation is intrinsically given by the seasonality of the climate. The real test is if the model has any predictive power on estimating the evaporative fraction AET/PET. If you normalize the AET on a daily timescale by the daily PET and then calculate the R2 and NSE, you probably get no explanation of variance. This can also be somewhat illustrated in table 6, if you add a column of RMSE in % of average daily AET, then you see that the RMSE is in the order of 50-80% of the daily AET (see attached table). In comparison most Remote sensing AET methods can, with calibration, achieve results in the order of RMSE of 25-30% of the daily mean AET.

Given the very little detail available on the AET model used (Feddes 1978) I can only

speculate, but perhaps the simulated SWC is not accurate enough at the critical moments when AET is limited by water availability, or the AET model is not appropriate or the climate data are poor. But overall I do not find the results on simulated daily AET encouraging. AN uncertainty analysis of teh different model components would be appropriate (see comment below)

Q: footprint analysis? EC footprint of 250 m radius is very large, what is the height of the EC mast?

Please explain the reasoning behind eq. 2 and 3?

L168: The Actual Transpiration is calculated using Feddes 1978 based on Tp and root density distribution. That must be a key component of this approach, please give more details on the application of the Feddes model.

L198-204: Optimized against which objective function? What was the calibration target? Which optimization algorithm (gradient based/global etc.) is used? That has to be clear up front? Also what was the result of the sensitivity analysis? Which type of sensitivity analysis, was it necessary to optimize all parameters? And why not calibrate all four layers simultaneous?

L220-224: It might be obvious, but please state clearly, which observation data the performance metrics are based on.

L230: How are the best defined, what are the weights and how was your combined objective function defined?

L265-266: Ofcause the upper layers are better, you calibrated them first and then kept them fixed while calibrating the lower layers, so they have had significantly more freedom in the optimization. Try to calibrate the lower first and then fix them and calibrate the upper, then you might get a different results.

L336: "the various ETa estimation techniques performed well." I disagree.

L337: "In fact, it is difficult to identify which is the clear solution if any." Please rephrase

Fig 9: How come the simulated values cannot go down to 0.20-0.25 for the Cosmic ray calibration, when that is possible for the TP calibrations?

Fig 11: The proposed method seems to not capture the interannual variability, try to plot the annual values of EC against simulated annual values in a scatterplot to see if there is any correlation on an annual basis?

Fig 13: You need to plot the daily obs vs. simulated AET in a scatterplot, the accumulated curves gives no indication of the performance of the daily model simulations! The bias of the Scatter plot will however give you the same information as the offset in accumulated values.

Table 6: Needs units.

I suggest resubmission of a new manuscript after major development of the inverse modelling and careful rethinking about the quality of the daily AET simulation results and reasons for the insufficient performance (AET model concept, upscaling, SWC simulations at critical stages, uncertainty in soil parameters, climate data etc.) . Here I would suggest some uncertainty analysis of the relative importance of these factors for the final AET results. E.g. how important are changes in soil parameters to the final result? And how important are the assumptions in the model (e.g. root depth, soil profile depths etc.)

Good luck

[Figure]

HESSD

Interactive
comment

Table 6: Assuming average daily obs. AET of 612/365 mm/day (from table 7)

| Location | R2 | MAE | RMSE | RMSE % of daily mean AET | NSE |
|---|---|---|---|---|---|
| TP1 | 0.652 | 0.696 | 1.062 | 63.3% | 0.618 |
| TP2 | 0.754 | 0.61 | 0.907 | 54.1% | 0.746 |
| TP3 | 0.751 | 0.601 | 0.904 | 53.9% | 0.728 |
| TP4 | 0.413 | 0.878 | 1.387 | 82.7% | 0.168 |
| CRNP | 0.499 | 0.787 | 1.259 | 75.1% | 0.349 |
**Fig. 1.**

---

## Referee Comment (RC2) · Anonymous Referee #2 · 20 Sep 2016

The manuscript describes an exploration into using ET derived using soil hydraulic parameters that are themselves inversely estimated from soil moisture measurements. The goal of the study is to validate additional data sources for LSMs. The manuscript is fairly well written, although further improvements can be made. While it is an interesting and required study, I do have a few concerns that I expect the authors to address before the manuscript can be accepted for publication.

P6, L114-119: Mention the instrument height above canopy for the EC tower. This would serve as a reference to validate your claim of the footprint size.

[Figure]

P7, L138-139: The reference to integration of CRNP data into the NOAH LSM seems extraneous here, and would be better deleted.

P7, L141-142: No numbers are given for the footprint size of the EC tower. So there's no way for the reader to decide if this assumption is valid or not. Further, with the assumption made, a discussion on the implications of this assumption later in the manuscript would be a good addition.

P8, L163: Please provide references to the Beer's law.

P8, L167: It may be better to mention that the LAI was described in the previous or study area section, rather than "above".

P8, L168: A brief description of how the Feddes model makes use of the potential transpiration and the root density distribution is necessary. Further, no details of the root density used in the study are given, which should be rectified.

P10, L199-205: What were the objective functions and methodology used to optimize these parameters? No description of any sort is provided, which makes it very difficult to assess the applicability.

P11, L223: R-squared has a name. It is called the Coefficient of Determination. Also, while the other metrics are described in equations, R-squared is not.

P12, L230: What about R-squared?

P12, L236: This may be a matter of semantics, but I feel that the subsection is better titled as "Vadoze Zone Inverse Modeling Results". You are performing inverse modeling of the vadose zone, not modeling of the inverse vadose zone.

P12, L238/239/250: Figures 4 and 7 are interchanged. Fig. 4 shows the annual precipitation, and fig 7 shows the temporal evolution of daily SWC.

P12, L239: Not so clear. It may be good to mention that the large standard deviation values show this. Also, I was surprised to see that the upper layers had smaller

SD values than the deeper layers! As the authors themselves mention elsewhere, the soil moisture variability is expected to reduce with depth. Any discussion on this phenomenon would be welcome.

P13, L272-273: Based on the numbers in Table 3, I am not sure the data are "fairly well matched". R-squared < 0.1 in the validation period (and < 0.4) in the calibration period), along with a negative NSE, tells me that the model and observation were not behaving alike. Maybe addition of distribution-level metrics could help bring out the relationship (if any) between the two better.

Also, here, and through the rest of the discussion, the authors use terms such as "fairly well matched" or "performed well" or similar language. These are highly subjective terms, and no analyses of numbers are provided to support these statements. It is necessary to establish at the beginning of the section what the authors consider as a "good" or "fairly good" etc., performance means in terms of absolute numbers. While the performance metrics are provided in the tables, no discussion is made regarding them and the reasoning for considering a particular statistic good.

P14, L282: How do these soil hydraulic parameters obtained from the inverse estimation compare with the textures used in the optimization? Further, while you mention earlier in the text that 6 different soil textures were used in the optimization, you omit mentioning which textures they are.

P14, L289: Provide a reference or hyperlink to the Web Soil Survey Data.

P15, L315: The infiltration rate in fine textured soil is lower, leading to higher surface runoff, as the authors mention. However, the water holding capacity of such soils is higher than coarse soils, leading to higher stored volume. I think a better argument here may be that the plant/root would have to overcome higher pressures to extract water from the fine soil, thus leading to lower ET.

P16, L330: Do you mean Figures 11 and 12 here? Figure 11 is never discussed in the

entire manuscript.

P16, L330-334: generally, the phenomenon of roots extracting water from deeper layers is seen in more mature vegetation such as trees, and not in seasonal agricultural crops. Also, even accepting that the plants may be drawing from layers deeper than the model domain, the phenomenon should not be so apparent in the clayey soils (TP4). A clayey soil restricts root penetration, and usually a shallow root depth is seen in such soils.

P16, L337: Clear solution to what?

Figures and Tables: I feel that, overall, the number of figures and tables can be reduced.

As mentioned earlier, Figures 4 and 7 are interchanged.

Figures 5 and 6: Keep any one of these two. No extra information is extracted by having two figures showing the same information here.

Figure 10: This can be merged with fig. 1.

Figure 11: This figure is never discussed in the text. Figure 12: This could be merged into fig. 11 as another panel. Also, in the text, this figure is discussed after fig. 13.

Table 1: These numbers can be discussed in the text instead of adding a single row table. As mentioned in an earlier comment, almost none of the numbers from the tables are discussed in context.

Table 3: Can be merged with tab. 2.

Table 7: There is no need for this table. The numbers can be mentioned in figs. 11 and 12. That would also make those figures easier to interpret.

Based on the above comments, I recommend that the authors be given an opportunity to make major revisions in the manuscript before resubmission.

[Figure]

Technical comments:

P4, L75: Should read as "... hyper-resolution LSM grid cells..." P5, L93: Check the spelling of the name "Simunek". P7, L135: "The CRNP measurement depth..." P7, L147: "... explained in detail by ..." P9, L176: "... GDD approximately 60-70..." P10, L195: The abbreviation TP has not been established earlier. P10, L198: the parameter "l" should be in lower case. P13, L262: "... criteria at TP locations..." P15, L302: "... inverse VZM modeling..." VZM already includes model. P16, L333: "VZM model" Same as above. References: Ensure uniform formatting of all the bibliography. Some end in page numbers, some in years, some in journal names, and some in volumes/issues. Table 4, Column 8: Use lower case "l" for tortuosity. Table 5, Column 6: Hectares in Field.

---

## Referee Comment (RC3) · Anonymous Referee #3 · 25 Sep 2016

Authors estimated field scale evapotranspiration (ET) by calibrating a 1D unsaturated zone model (HYDRUS-1D) using soil water content measurements, and compared simulated ET with observed ET from an eddy covariance tower. The HYDRUS-1D soil hydraulic parameters were calibrated using daily soil water content measurements from four theta monitoring probes at multiple depths and one cosmic ray neutron probe. While this is an interesting study, the novelty of the current study is not clear. Based on presented results, large differences exist between simulated ET and eddy covariance data and results of soil moisture simulations are not entirely satisfactory given the negative NSE during calibration and small coefficient of determination for soil moisture

simulations at certain depths. In particular, authors have not discussed the implications of their results and what can be done to improve model estimation. While the focus of the inverse modelling was on soil hydraulic parameters estimation, the study can benefit from a detailed model sensitivity experiment to soil hydraulic and root growth function model parameters. I suggest authors to perform a detailed uncertainty estimation approach to identify the sources of errors (model input, parameters, or model structure) in ET and soil water content estimates. This can help to identify why the model did not perform well in some cases and how authors can improve their results.

1. Introduction, the rational and implications of the current study are not entirely clear. I suggest authors outline the main objectives of their study and discuss how their results advance our understanding of ET estimation using unsaturated zone models. It is not clear whether authors try to develop a benchmark for soil moisture or ET estimation or how their soil hydraulic parameter estimation can help parametrize hyper-resolution land surface models? These are the ideas that are discussed in the Introduction but their links with the current study are not clear.

2. Section 2.2.1. It seems authors have used a different growth root model compared to the HYDRUS-1D root growth model for annual vegetation. Have authors performed any experiments to assess how the results of the two root growth models compare?

3. Section 2.2.1. It will be very useful if authors can report Kc parameters and root growth model parameters as they can impact the results of ET estimation.

4. Section 2.2.2. Additional details regarding the inverse modelling algorithm and an objective function that is used for parameter estimation are required.

5. Section 2.2.2. Line 206- Can authors provide further details about initial soil hydraulic parameters that they used in the modelling experiment? Did they use soil hydraulic parameters based on soil texture class information? Similarly, authors used the same parameter bounds for model calibration for all soil texture classes. It will be useful if authors can incorporate the soil texture information to define priors and initial

parameter values.

6. Section 2.2.2. Why homogeneous soil type was used for simulating water content for the Cosmos-Ray neutron probe while for the Theta probes variability in vertical hydraulic conductivity is considered?

7. Why the spin-up period is varied between the inverse modelling approach and the forward model? What criteria authors used to define model spin-up?

8. Table 2-Why negative NSE is obtained during calibration period particularly in deeper soil layers? Even R2 values are pretty small for a VZM model that is calibrated to observations. Can authors describe the reasons for this mismatch? Similarly results of soil moisture simulation are not satisfactory for the CRNP calibration based on Table 3.

9. Authors indicate that inverse modelling based on CRNP data is most useful during the periods that soil evaporation is dominant. Can authors further explain why that is the case? One would expect that CRNP should provide better estimate of ET as its footprint is likely to overlap the EC tower footprint.

10. Section 3.2. Authors relate variability in performance of the model in ET simulation to variability in soil texture. However, one important information that is missing is vegetation type at the location of the probes and the EC tower footprint scale. Perhaps, authors should combine ET estimates from multiple probes to estimate ET at a field scale.

11. It will be useful if authors can provide information about deep drainage from model simulations at multiple locations.

Minor comments: Figures 1 and 2 can be combined in one Figure.

Line 166- Extinction

Line 238-Please revise the Figure number to 7.

Figure 5- Can authors describe the reason for large differences between the spatially averaged TP and CRNP by the end of year 2014?

[Figure]

---

## Author Comment (AC1) · 11 Oct 2016

The study addresses an important issue of enlarging the data sets available for LSM validation by estimating AET from SWC measurements. Also the underlying idea that recharge and AET data are generally more valuable to society than SWC alone justifies this field of research. The manuscript is very well written. However:

1. The inverse methodology description is very weak. There is no description of which search method is used! What is the combined objective function? A detailed sensitivity analysis has to be given, especially in light of the mentioned problems of equifinality. It is extremely unlikely that all 24 parameters are sensitive and justify optimization. Also inverse modelling offers the opportunity to provide the reader with an estimate of the confidence intervals for each estimated parameter, which will also reveal the sensitivity and associated uncertainty.

   *Thank you so much for your comments. The central theme of the paper was to employ a standard publicity available model to test our hypothesis, not to devise new algorithms for inversion, and that is why we did not get into the inverse modeling details in great depth. As it was mentioned in the paper, more description about inverse modeling can be found in Mualem (1976), van Genuchten (1980), and Turkeltaub et al. (2015).*

   *Moreover, Wang et al (2009) have done a detailed sensitivity analysis of groundwater recharge and evapotranspiration for soil hydraulic parameters in a single layer. We respect your concerns and have undertaken a sensitivity analysis of all 4 layers (24 parameters) extending the original work of Wang et al (2009).*

   *Wang, T., V. A. Zlotnik, J. Simunek, and M. G. Schaap. 2009. Using pedotransfer functions in vadose zone models for estimating groundwater recharge in semiarid regions. Water Resources Research 45: 12. doi:10.1029/2008wr006903.*

2. The results of simulated SWC seems to be reasonable from a SWC perspective, but it's important to also address the certainty/robustness and likelihood of the estimated soil parameters. Are they random parameter picks from an equifinal problem or are they physically reasonable and do their mutual differences fit into field/lab measurements (I assume soil samples exists from the sites)? The author has attempted to validate the spatial distribution of the estimated soil parameters based on a soil map, which is highly appreciated. However, it would have been interesting to utilize this information for regionalizing the soil parameters and thereby limiting the number of free parameters in the calibration. Likewise, the soil map could have been used to upscale the AET simulations to the field scale by including the soil map instead of a simple average of the four points.

*Thank you for the suggestion. Upscaling the AET by the SSURGO soil map will be interesting. We also note that Professor Franz uses a range of hydrogeophysical mapping techniques (i.e. electromagnetic induction, cosmic-ray neutron rover) to understand soil patterns and properties. We have mapped this site several times and will consider adding some of the maps to this manuscript or a companion manuscript.*

3. The results of the AET simulations seem to be very poor. I miss a critical view on the results regarding lacking ability to simulate even inter-annual variability (fig 11) and perhaps more importantly the apparently complete lack of predictive capability on the daily scale. The performance metrics in Table 6 indicate good $R^2$ and NSE, but that correlation is intrinsically given by the seasonality of the climate. The real test is if the model has any predictive power on estimating the evaporative fraction AET/PET. If you normalize the AET on a daily timescale by the daily PET and then calculate the $R^2$ and NSE, you probably get no explanation of variance. This can also be somewhat illustrated in table 6, if you add a column of RMSE in % of average daily AET, then you see that the RMSE is in the order of 50-80% of the daily AET (see attached table). In comparison most Remote sensing AET methods can, with calibration, achieve results in the order of RMSE of 25-30% of the daily mean AET.

*We compared the results with EC measured ET in this study just as a simple comparison as there was no other relatively accurate measured ET data available in the study area. As it was mentioned in the paper there are always different uncertainties involve in the Eddy-Covariance (EC) measurements. EC measurements can be bias by up to 20% or even more. Considering this we cannot easily say since the simulated ET values are not perfectly matched with EC ET measured data that "the AET simulations seem to be very poor".*

*Your suggestions are appreciated and if we had access to more accurate measured ET data, like Lysimeter measured ET, we could investigate such analysis. Because of the nature of EC ET measurements (which is not based on Kc values, but instead based on the flux measurements) such comparison may not be useful. As an example obtained Kc values from EC (2007-2012) are shown below. According to the graph, most of the times during mid-growing season, obtained Kc values from EC are less than 1 (we usually expect to have values of 1-1.2 during the mid-growth season). The average EC Kc value during July and August (2007-2012) is 0.81 with a minimum average Kc value of 0.58 in 2012 and maximum Kc value of 0.99 in 2011. On the other hand, sometimes Kc values exceed 4 while in the real world such Kc values do not exist. In addition, Kc values do not usually change suddenly during growing season and it is rarely possible to have Kc value of 1 in one day and Kc value of 0.4 for the next day, but according to the graphs in some of the days we can see this case in the EC Kc values. The inherent noise seen in Kc makes this comparison challenging without temporal smoothing.*

[Figure]

4. Given the very little detail available on the AET model used (Feddes 1978) I can only speculate, but perhaps the simulated SWC is not accurate enough at the critical moments when AET is limited by water availability, or the AET model is not appropriate or the climate data are poor. But overall I do not find the results on simulated daily AET encouraging. An uncertainty analysis of the different model components would be appropriate (see comment below).

*We appreciate the reviewer's thoughts, but most of the comments are made based on the apparent difference between the EC measured ET and simulated ET. The Hydrus model is a widely used method based on a solution to the Richards Equation. The Mead Site 3 flux tower is a long standing Ameriflux tower and continues to be a part of the core network. In order to address the comments, we will perform a sensitivity analysis of all 24 soil hydraulic properties building on Wang et al. (2009). A full sensitivity analysis of the root model parameters is beyond the current scope of the paper and we refer the reviewer to Guswa (2012).*

*Guswa, A. J. 2012. Canopy vs. Roots: Production and Destruction of Variability in Soil Moisture and Hydrologic Fluxes. Vadose Zone Journal 11:3. doi:10.2136/vzj2011.0159.*

Q: footprint analysis? EC footprint of 250 m radius is very large, what is the height of the EC mast?

*The height on the EC mast varies with crop height. According to Suyker et al. (2004):*

*"To have sufficient upwind fetch (in all directions) representative of the cropping system being studied, eddy covariance sensors were mounted at 3.0 m above the ground while the canopy was shorter than 1.0 m, and later moved to a height of 6.2 m until harvest."*

*The footprint of the tower will there change over the season, ~100 times the tower height. This is a long running Ameriflux site and the variable footprint is a part of the method and its inherent uncertainty.*

*Suyker, A. E., S. B. Verma, G. G. Burba, T. J. Arkebauer, D. T. Walters, and K. G. Hubbard. 2004. Growing season carbon dioxide exchange in irrigated and rainfed maize. Agric. For. Meteorol. 124:1-2: 1-13. doi:10.1016/j.agrformet.2004.01.011.*

5. Please explain the reasoning behind eq. 2 and 3?

   *We needed to introduce potential evaporation (Ep) and potential transpiration (Tp) values to the Hydrus model. By using Beer's law we were able to divide ETp to Ep and Tp. Based on LAI values, with equation 2 we can calculate the Ep values and then by having the Ep value we can use equation 3 to calculate the Tp values. More information can be found in Šimunek et al, (2013).*

   *Šimunek, J., Šejna, M., Saito, H., Sakai, M., van Genuchten, M.T. (2013). The HYDRUS-1D Software Package for Simulating the One-Dimensional Movement of Water, Heat, and Multiple Solutes in Variably-Saturated Media, Version 4.17. Department of Environmental Sciences, University of California Riverside, Riverside, California, USA, 307 pp.*

6. L168: The Actual Transpiration is calculated using Feddes 1978 based on Tp and root density distribution. That must be a key component of this approach, please give more details on the application of the Feddes model.

   *According to Šimunek et al, (2013), S is a sink term and has been defined as plant water uptake:*

$$S(h) = \alpha(h)S_p$$

   *where $\alpha(h)$ is a dimensionless function varies between 0 and 1 depending upon soil water pressure head and $S_p$ is the potential water uptake rate assumed to be equal to Tp. More information can be found in Šimunek et al, (2013), and Wang et al. (2016). We will add a better description to the manuscript. See Guswa (2012) for a more in depth look.*

7. L198-204: Optimized against which objective function? What was the calibration target? Which optimization algorithm (gradient based/global etc.) is used? That has to be clear up front? Also what was the result of the sensitivity analysis? Which type of sensitivity analysis, was it necessary to optimize all parameters? And why not calibrate all four layers simultaneous?

*We could not optimize all the layers simultaneously because the maximum number of parameters that we can be optimized by the Hydrus-1D model is 15. We have followed the same procedure as Turkeltaub et al. (2015) and Wang et al. (2015, 2016). We used RMSE as our objective function and will clarify this more in the manuscript. Finally, a sensitivity analysis of all 24 parameters will be presented.*

8. L220-224: It might be obvious, but please state clearly, which observation data the performance metrics are based on.

   *The performance metrics are based on the soil water content data and it will be added to the modified manuscript.*

9. L230: How are the best defined, what are the weights and how was your combined objective function defined?

   *We chose the selected optimized sets of soil parameters values based on the RMSE but the other objective functions were performed in order to double check the optimization process. We will clearly state in the modified manuscript that the soil hydraulic parameters were chosen based on the RMSE.*

10. L265-266: Of course the upper layers are better you calibrated them first and then kept them fixed while calibrating the lower layers, so they have had significantly more freedom in the optimization. Try to calibrate the lower first and then fix them and calibrate the upper, then you might get different results.

    *The Hydrus-1D can just optimize up to 15 parameters simultaneously and we decided to optimize the upper 2 layers first and then the 2 lower layers. The SWC data in the 2 upper layers has more dynamics than the 2 lower layers. Again a sensitivity analysis of all 24 parameters will be presented. Interestingly, preliminary results indicate the deepest layer n value is the most sensitive.*

11. L336: "the various ETa estimation techniques performed well." I disagree.

    *Each method, measurements, inverse modelling, are prone to uncertainties. Unfortunately, no true ETa estimate exists at the field scale. We will try to soften the language here.*

12. L337: "In fact, it is difficult to identify which is the clear solution if any." Please rephrase.

    *Thank you, we will rephrase it.*

13. Fig 9: How come the simulated values cannot go down to 0.20-0.25 for the Cosmic ray calibration, when that is possible for the TP calibrations?

*We pointed out in the paper during the growing season when crops extract water from deeper soil layers comparison between simulated and observed values deteriorates due to the fact that the CRNP observational depth is limited to near surface layers (~20 cm).*

14. Fig 11: The proposed method seems to not capture the inter-annual variability, try to plot the annual values of EC against simulated annual values in a scatterplot to see if there is any correlation on an annual basis?

*Thank you for your suggestion. We will try to plot the annual measured EC ET values versus simulated annual values in a scatterplot to see if that seems more informative.*

15. Fig 13: You need to plot the daily obs vs. simulated AET in a scatterplot, the accumulated curves gives no indication of the performance of the daily model simulations! The bias of the Scatter plot will however give you the same information as the offset in accumulated values.

*We have investigated the plots and will provide the 1:1 in the full response to reviewers.*

16. Table 6: Needs units.

*Thank you, we will add them to the table.*

17. I suggest resubmission of a new manuscript after major development of the inverse modelling and careful rethinking about the quality of the daily AET simulation results and reasons for the insufficient performance (AET model concept, upscaling, SWC simulations at critical stages, uncertainty in soil parameters, climate data etc.). Here I would suggest some uncertainty analysis of the relative importance of these factors for the final AET results. E.g. how important are changes in soil parameters to the final result? And how important are the assumptions in the model (e.g. root depth, soil profile depths etc.).

*Thank you for the very insightful comments. Obviously the reviewer is well versed in this type of analysis. We will try our best to resolve the above issues. A sensitivity analysis of all 24 soil parameters in combination with the work of Wang et al. (2009) will be instructive.*

Good luck

---

## Author Comment (AC2) · 11 Oct 2016

The manuscript describes an exploration into using ET derived using soil hydraulic parameters that are themselves inversely estimated from soil moisture measurements. The goal of the study is to validate additional data sources for LSMs. The manuscript is fairly well written, although further improvements can be made. While it is an interesting and required study, I do have a few concerns that I expect the authors to address before the manuscript can be accepted for publication.

*Thank you for the very insightful comments.*

P6, L114-119: Mention the instrument height above canopy for the EC tower. This would serve as a reference to validate your claim of the footprint size.

*According to Suyker 2004, the height of EC tower is 6 meter during the growing season and 3 meter before and after the growing season when nothing planting. We will add these details.*

P7, L138-139: The reference to integration of CRNP data into the NOAH LSM seems extraneous here, and would be better deleted.

*Thank you we will delete it.*

P7, L141-142: No numbers are given for the footprint size of the EC tower. So there's no way for the reader to decide if this assumption is valid or not. Further, with the assumption made, a discussion on the implications of this assumption later in the manuscript would be a good addition.

*Thank you we will add that to the modified manuscript.*

P8, L163: Please provide references to the Beer's law.

*Thank you we will add that to the modified manuscript.*

P8, L167: It may be better to mention that the LAI was described in the previous or study area section, rather than "above".

*Thank you we will change that.*

P8, L168: A brief description of how the Feddes model makes use of the potential transpiration and the root density distribution is necessary. Further, no details of the root density used in the study are given, which should be rectified.

*Thank you we will add that to the modified manuscript. A full sensitivity analysis of the root model parameters is beyond the current scope of the paper and we refer the reviewer to Guswa (2012).*

*Guswa, A. J. 2012. Canopy vs. Roots: Production and Destruction of Variability in Soil Moisture and Hydrologic Fluxes. Vadose Zone Journal 11:3. doi:10.2136/vzj2011.0159.*

P10, L199-205: What were the objective functions and methodology used to optimize these parameters? No description of any sort is provided, which makes it very difficult to assess the applicability.

*We could not optimize all the layers simultaneously because the maximum number of parameters that we can be optimized by the Hydrus-1D model is 15. We have followed the same procedure as Turkeltaub et al. (2015) and Wang et al. (2015, 2016). We used RMSE as our objective function and will clarify this more in the manuscript. Finally, a sensitivity analysis of all 24 parameters will be presented.*

P11, L223: R-squared has a name. It is called the Coefficient of Determination. Also, while the other metrics are described in equations, R-squared is not.

*Thank you we will change it in the modified manuscript.*

P12, L230: What about R-squared?

*In fact the primary objective function that was used to find the best sets of soil hydraulic parameters was RMSE and the others just were used to double check the optimization process. In the modified manuscript we will mention that we use RMSE values to choose the best set of the soil parameters. However, you are right and we should (and we will) add and name Coefficient of Determination (R-squared) as one of the objectives functions which were used for more investigation.*

P12, L236: This may be a matter of semantics, but I feel that the subsection is better titled as "Vadoze Zone Inverse Modeling Results". You are performing inverse modeling of the vadose zone, not modeling of the inverse vadose zone.

*Thank you, agreed. We will change it in the modified manuscript.*

P12, L238/239/250: Figures 4 and 7 are interchanged. Fig. 4 shows the annual precipitation, and fig 7 shows the temporal evolution of daily SWC.

*Thank you, we will correct it in the modified manuscript.*

P12, L239: Not so clear. It may be good to mention that the large standard deviation values show this. Also, I was surprised to see that the upper layers had smaller SD values than the deeper layers! As the authors themselves mention elsewhere, the soil moisture variability is expected to reduce with depth. Any discussion on this phenomenon would be welcome.

*You are right we should say "according to the standard deviation value SWC varies considerably across the site, particularly during the growing especially in the deeper layers". When we say soil moisture variability is expected to reduce with depth we meant soil moisture variability expected to reduce with respect to time in each location alone not soil moisture variability in one location versus the other locations. We will clarify this in the revisions.*

P13, L272-273: Based on the numbers in Table 3, I am not sure the data are "fairly well matched". R-squared < 0.1 in the validation period (and < 0.4) in the calibration period), along with a negative NSE, tells me that the model and observation were not behaving alike. Maybe addition of distribution-level metrics could help bring out the relationship (if any) between the two better.

Also, here, and through the rest of the discussion, the authors use terms such as "fairly well matched" or "performed well" or similar language. These are highly subjective terms, and no analyses of numbers are provided to support these statements. It is necessary to establish at the beginning of the section what the authors consider as a "good" or "fairly good" etc., performance means in terms of absolute numbers. While the performance metrics are provided in the tables, no discussion is made regarding them and the reasoning for considering a particular statistic good.

*In this study we tried to optimize soil hydraulic parameters based on the simulated SWC and observed SWC. RMSE was chosen as the main objective function to select the best sets of soil hydraulic parameters.*

*Thank you for your suggestion, we will add a section at the beginning and describe performance means in terms of absolute numbers for clarity.*

P14, L282: How do these soil hydraulic parameters obtained from the inverse estimation compare with the textures used in the optimization? Further, while you mention earlier in the text that 6 different soil textures were used in the optimization, you omit mentioning which textures they are.

*The soil texture data used in the optimization were just the model default and we used 6 sets of them in order to find the best sets of soil hydraulic parameters in the site based on the observed SWC. We will add this description to the text. Since we start with all 6 guesses not sure what this suggestion would accomplish?*

P14, L289: Provide a reference or hyperlink to the Web Soil Survey Data.

*We will add reference.*

P15, L315: The infiltration rate in fine textured soil is lower, leading to higher surface runoff, as the authors mention. However, the water holding capacity of such soils is higher than coarse soils, leading to higher stored volume. I think a better argument here may be that the plant/root would have to overcome higher pressures to extract water from the fine soil, thus leading to lower ET.

*Agreed. Thank you for the comment, we will add suggestion to modified manuscript.*

P16, L330: Do you mean Figures 11 and 12 here? Figure 11 is never discussed in the entire manuscript.

*Yes. We meant Figure 12 and 13, but we will check out how we have missed Figure 11 in the manuscript.*

P16, L330-334: generally, the phenomenon of roots extracting water from deeper layers is seen in more mature vegetation such as trees, and not in seasonal agricultural crops. Also, even accepting that the plants may be drawing from layers deeper than the model domain, the phenomenon should not be so apparent in the clayey soils (TP4). A clayey soil restricts root penetration, and usually a shallow root depth is seen in such soils.

*Those were our initial thoughts, but conversations with the site PI agronomists suggest water extraction up to 2 m, even in clayey soils! This is based on SWC readings from neutron access tubes in the surrounding fields part of the larger University of Nebraska Mead Extension Center.  Root water uptake is very complex and we refer the reader to Guswa 2012 for a more in depth discussion. Also in TP4 location we said we expect to have clayey soils and Web Soil Survey Data confirms our results.  We note that this conclusion is based on our simulation results and Web Soil Survey Data which only provides information for the upper soil layer not the deeper layers. Clearly investigation in root water uptake is an area that deserves more attention in LSMs, even in homogeneous annual crops. We are investigating spatial root and soil water interaction using hydrogeophysical mapping techniques in Prof. Franz's laboratory.*

*Guswa, A. J. 2012. Canopy vs. Roots: Production and Destruction of Variability in Soil Moisture and Hydrologic Fluxes. Vadose Zone Journal  11:3. doi:10.2136/vzj2011.0159.*

P16, L337: Clear solution to what?

*Thank you, we will rephrase it.*

Figures and Tables: I feel that, overall, the number of figures and tables can be reduced. As mentioned earlier, Figures 4 and 7 are interchanged.

*Thank you,  we will correct that.*

Figures 5 and 6: Keep any one of these two. No extra information is extracted by having two figures showing the same information here.

*Thank you, we will remove one of them.*

Figure 10: This can be merged with fig. 1.

*Thank you. We will consider this.*

Figure 11: This figure is never discussed in the text. Figure 12: This could be merged into fig. 11 as another panel. Also, in the text, this figure is discussed after fig. 13.

*Thank you. We will check that out we should have missed it, and will merge Figures 11 and 12 to one figure. We check out and if that is the case we may need to change figures numbers.*

Table 1: These numbers can be discussed in the text instead of adding a single row table. As mentioned in an earlier comment, almost none of the numbers from the tables are discussed in context.

*Thank you. We will consider this.*

Table 3: Can be merged with tab. 2.

*Thank you. We will consider this.*

Table 7: There is no need for this table. The numbers can be mentioned in figs. 11 and 12. That would also make those figures easier to interpret.

*Thank you. We will consider this.*

Based on the above comments, I recommend that the authors be given an opportunity to make major revisions in the manuscript before resubmission.

Technical comments:
P4, L75: Should read as "… hyper-resolution LSM grid cells…" P5, L93: Check the spelling of the name "Simunek". P7, L135: "The CRNP measurement depth…" P7, L147: "… explained in detail by …" P9, L176: "… GDD approximately 60-70…" P10, L195: The abbreviation TP has not been established earlier. P10, L198: the parameter "l" should be in lower case. P13, L262: "… criteria at TP locations…" P15, L302: "… inverse VZM modeling…" VZM already includes model. P16, L333: "VZM model" Same as above. References: Ensure uniform formatting of all the bibliography. Some end in page numbers, some in years, some in journal names, and some in volumes/ issues. Table 4, Column 8: Use lower case "l" for tortuosity. Table 5, Column 6: Hectares in Field.

*Thank you,  we will make corrections.*

---

## Author Comment (AC3) · 11 Oct 2016

Authors estimated field scale evapotranspiration (ET) by calibrating a 1D unsaturated zone model (HYDRUS-1D) using soil water content measurements, and compared simulated ET with observed ET from an eddy covariance tower. The HYDRUS-1D soil hydraulic parameters were calibrated using daily soil water content measurements from four theta monitoring probes at multiple depths and one cosmic ray neutron probe. While this is an interesting study, the novelty of the current study is not clear. Based on presented results, large differences exist between simulated ET and eddy covariance data and results of soil moisture simulations are not entirely satisfactory given the negative NSE during calibration and small coefficient of determination for soil moisture simulations at certain depths. In particular, authors have not discussed the implications of their results and what can be done to improve model estimation. While the focus of the inverse modelling was on soil hydraulic parameters estimation, the study can benefit from a detailed model sensitivity experiment to soil hydraulic and root growth function model parameters. I suggest authors to perform a detailed uncertainty estimation approach to identify the sources of errors (model input, parameters, or model structure) in ET and soil water content estimates. This can help to identify why the model did not perform well in some cases and how authors can improve their results.

*Thank you for your comments. The central theme of the paper was to employ a standard publicity available model to test our hypothesis, not to devise new algorithms for inversion, and that is why we did not get into the inverse modeling details in great depth. As it was mentioned in the paper, more description about inverse modeling can be found in Mualem (1976), van Genuchten (1980), and Turkeltaub et al. (2015).*

*Moreover, Wang et al (2009) have done a detailed sensitivity analysis of groundwater recharge and evapotranspiration for soil hydraulic parameters in a single layer. We respect your concerns and have undertaken a sensitivity analysis of all 4 layers (24 parameters) extending the original work of Wang et al (2009). A full sensitivity analysis of the root model parameters is beyond the current scope of the paper and we refer the reviewer to Guswa (2012).*

*Guswa, A. J. 2012. Canopy vs. Roots: Production and Destruction of Variability in Soil Moisture and Hydrologic Fluxes. Vadose Zone Journal 11:3. doi:10.2136/vzj2011.0159.*

*Wang, T., V. A. Zlotnik, J. Simunek, and M. G. Schaap. 2009. Using pedotransfer functions in vadose zone models for estimating groundwater recharge in semiarid regions. Water Resources Research 45: 12. doi:10.1029/2008wr006903.*

1. Introduction, the rational and implications of the current study are not entirely clear. I suggest authors outline the main objectives of their study and discuss how their results advance our understanding of ET estimation using unsaturated zone models. It is not clear whether authors try to develop a benchmark for soil moisture or ET estimation or how their soil hydraulic parameter estimation can help parametrize hyper-resolution land surface

models? These are the ideas that are discussed in the Introduction but their links with the current study are not clear.

*Thank you for the comments. We will seek to improve rational of manuscript in the introduction.*

2. Section 2.2.1. It seems authors have used a different growth root model compared to the HYDRUS-1D root growth model for annual vegetation. Have authors performed any experiments to assess how the results of the two root growth models compare?

*Since we had annual cultivation rotation between soybean and maize we had to introduce the root depth to the model and we could not use the default values inside the model. Likewise, as default values were constant and cannot be changed for different type of crops in different years during the simulation, we were not able to compare the models. This parameterization is not available in the standard HYDRUS package and a limitation of using it with crop rotations. We wanted to keep intact the cropping history to minimize impact on SWC between years. Clearly the topic of root water uptake deserves more investigation.*

3. Section 2.2.1. It will be very useful if authors can report Kc parameters and root growth model parameters as they can impact the results of ET estimation.

*As it was mentioned in the manuscripts the suggested Kc values by Allen et al. (1998) for maize and soybean were used. For root growth model the maximum root depth assumed equal to 150 cm for maize and 120 cm for soybean. In addition, GDD was calculated by mentioned equation using Tmax, Tmin, and Tbase. We will try and clarify in revisions.*

4. Section 2.2.2. Additional details regarding the inverse modelling algorithm and an objective function that is used for parameter estimation are required.

*The maximum number of parameters that we can be optimized by the Hydrus-1D model is 15. We have followed the same procedure as Turkeltaub et al. (2015) and Wang et al. (2015, 2016). We used RMSE as our objective function and will clarify this more in the manuscript. Finally, a sensitivity analysis of all 24 parameters will be presented in the revisions.*

5. Section 2.2.2. Line 206- Can authors provide further details about initial soil hydraulic parameters that they used in the modelling experiment? Did they use soil hydraulic parameters based on soil texture class information? Similarly, authors used the same parameter bounds for model calibration for all soil texture classes. It will be useful if authors can incorporate the soil texture information to define priors and initial parameter values.

*The initial values were just the default values in the Hydrus-1D model which are based on the different soil types. Agreed, priors could be used with pedotransfer functions to improve results. Unfortunately, the connection between hydrologic fluxes and soil texture classes is unclear (Groenendyk et al. 2015). This work continues on that disconnection and will be highlighted more in the revisions.*

*Groenendyk, D. G., T. P. A. Ferre, K. R. Thorp, and A. K. Rice. 2015. Hydrologic-Process-Based Soil Texture Classifications for Improved Visualization of Landscape Function. PLoS One 10:6: 17. doi:10.1371/journal.pone.0131299.*

6. Section 2.2.2. Why homogeneous soil type was used for simulating water content for the Cosmos-Ray neutron probe while for the Theta probes variability in vertical hydraulic conductivity is considered?

   *As a first cut we used a single layer. Since the CRNP only sees the top 20 cm we wanted to see how well it could or not reproduce ETa values. Clearly more investigation is needed about the use of CRNP to estimate ETa.*

7. Why the spin-up period is varied between the inverse modelling approach and the forward model? What criteria authors used to define model spin-up?

   *We have followed the same procedure as Wang et al. (2015, 2016) for model spin up. We will clarify this in the text.*

8. Table 2-Why negative NSE is obtained during calibration period particularly in deeper soil layers? Even R2 values are pretty small for a VZM model that is calibrated to observations. Can authors describe the reasons for this mismatch? Similarly results of soil moisture simulation are not satisfactory for the CRNP calibration based on Table 3.

   *We will add clarification to revisions.*

9. Authors indicate that inverse modelling based on CRNP data is most useful during the periods that soil evaporation is dominant. Can authors further explain why that is the case? One would expect that CRNP should provide better estimate of ET as its footprint is likely to overlap the EC tower footprint.

   *Since the CRNP only sees the top 20 cm we wanted to see how well it could or not reproduce ETa values. We hypothesize that at roots development into deeper layers and Transpiration becomes more important in the latent energy term the information content in the CRNP would diminish. Clearly this topic requires more investigation. We will add clarification to revisions.*

10. Section 3.2. Authors relate variability in performance of the model in ET simulation to variability in soil texture. However, one important information that is missing is vegetation type at the location of the probes and the EC tower footprint scale. Perhaps, authors should combine ET estimates from multiple probes to estimate ET at a field scale.

*Thank you for the suggestion. Based on reviewer 1, we will investigate if upscaling the AET by the SSURGO soil map will be useful. We also note that Professor Franz uses a range of hydrogeophysical mapping techniques (i.e. electromagnetic induction, cosmic-ray neutron rover) to understand soil patterns and properties. We have mapped this site several times and will consider adding some of the maps to this manuscript or a companion manuscript. We note the vegetation will be the same for all locations. Destructive vegetation sampling at each location is available from the site PIs to look at variability in the canopy.*

11. It will be useful if authors can provide information about deep drainage from model simulations at multiple locations.

*We will consider adding this to the manuscript.*

Minor comments: Figures 1 and 2 can be combined in one Figure.

12. Line 166- Extinction

*Thank you, we will correct it.*

13. Line 238-Please revise the Figure number to 7.

*Thank you, we will revise it.*

14. Figure 5- Can authors describe the reason for large differences between the spatially averaged TP and CRNP by the end of year 2014?

*We will consider this comment in the revised manuscript.*

---

## Author Response (AR1)

Dear Prof. McCabe,

We would like to thank you and the three reviewers for your time and excellent comments regarding our manuscript, titled "Feasibility analysis of using inverse modeling for estimating field-scale evapotranspiration in maize and soybean fields from soil water content monitoring networks". After careful analysis of all the comments, we have made extensive revisions to our manuscript. You can find our detailed responses to the reviewers' comments (shown in red italics) and the changes we made to the manuscript in the following sections. We have also included a marked up version of the original manuscript.

On the behalf of all coauthors, I hope that this revised version would meet the publication standard of Hydrology and Earth System Sciences (HESS) and inclusion in the Eric F. Wood special issue. Please let us know if there are more questions and comments about the manuscript.

Sincerely,

Prof. Trenton E. Franz

School of Natural Resources

University of Nebraska-Lincoln, USA

Reply to the editor:

*Thank you for the comments regarding our manuscript. Please see our detailed replies below.*

1. All of the reviewers have requested some details on the type and manner of inverse modeling. Given the importance of this element to your work, it would be helpful to see some additional methodological paragraphs on this, rather than just referencing previous publications.

   *Authors: Thank you for the suggestion. We have included more detail about the inverse methodology, which is more "off the shelf". We note that we use RMSE as our objective function to minimize in order to select parameters using the built in Hydrus software. We also have included other fitting metrics for completeness. Please see L231-234 for full details.*

   *L231-234: "With respect to the goodness-of-fit assessment, Root Mean Square Error (RMSE) between simulated and observed SWC was chosen as the objective function to minimize in order to estimate the soil hydraulic parameters. The built in optimization procedure in Hydrus-1D was used to perform parameter estimation."*

2. Some further comment and discussion on the quality of the reproduced evaporation is warranted. Accurately (inversely) modeling the ET is clearly non-trivial and there are a multitude of possible reasons that could affect its simulation beyond just issues to do with the eddy-covariance approach. Outlining these and providing some insights and guidance where possible (e.g. the assessment of parameter uncertainty and influence on ET response) would add considerable value to the manuscript.

   *Thank you for the suggestions. In order to investigate the key sources of error, as you suggested, we performed a set of preliminary sensitivity analysis experiments of effects of soil hydraulic parameters and plant root growth on the ETa and results are presented in part 3.3 (L383-L417) and figures 10 and 11. The preliminary sensitivity analysis on a number of key soil and plant parameters was very insightful and has improved the manuscript considerably. We also provide a description and a few key citations (Bastidas et al. 1999 and Rosolem et al. 2012) to undertake a more in depth sensitivity analysis in future work.*

3. Certainly there is no need to overstate whether the approaches accurately match the eddy-covariance data: if the retrieval is judged to be relatively poor, the work still presents useful findings – especially if these can be related to either the interpretive model used or some other reason (parameter uncertainty, equifinality issues, measurement limitations).

*Thank you for the comment. We tried to avoid overstating this in the manuscript and now provide a set of goodness of fit metrics based off RMSE. This is done for both soil water content (L278-L281) and ETa (L331-333).*

*L278-L281: "In this research we define RMSE values less than 0.03 $cm^3/cm^3$ between observed and simulated SWC values as well-matched and RMSE between 0.03 and 0.06 $cm^3/cm^3$ as fairly well-matched. We note the target error range of satellite SWC products (e.g. SMOS and SMAP) is less than 0.04 $cm^3/cm^3$ (Entekhabi et al., 2010).*

*L331-333: "In this research we consider RMSE values less than 1 mm/day between observed and simulated $ET_a$ values as well-matched and RMSE values between 1 and 1.2 as fairly well-matched (Figure 9 and Table 6)."*

4.  Related to this, the paper would benefit by adding some detail of the evaporation sub-model, perhaps placing this in the context of other approaches that can be employed to estimate evaporation using the data that you have available (including the met data that would have been collected by the EC system). If it is a simplistic approach, perhaps it is unreasonable to expect an accurate reproduction?

*Thank you for the comment. Here we have focused on using two different data sources and HYDRUS to estimate evaporation and associated parameters. We found that the CRNP does a reasonable job (based off SWC and ETa RMSE scores and fit criteria) to constrain HYDRUS in the top layer. From the sensitivity analysis it seems constraining the soil hydraulic parameters n and alpha are critical for the top layer, indicating the CRNP may be useful in estimating evaporation, particularly when transpiration is relatively small. For more simple models we suggest that other widely used remote sensing and crop models may benefit from a constrained evaporation estimate from CRNP (L397-399).*

*L397-399: "Moreover, the CRNP may be useful in helping constrain and parameterize soil hydraulic functions in simpler evaporation models used in remote sensing (c.f. Allen et al. 2007) or crop modeling (c.f. Allen et al. 1998)."*

5.  Where possible, reduce and merge figures and tables, only maintaining those that are directly relevant to the material being presented.

*Thank you for the comment. We tried to reduce and merge the figures as suggested by the reviewers. Reduced figures from 13 to 11 but added sensitivity analysis with same number of tables with new analyses suggested by reviewers.*

Replies to Anonymous Reviewer #1

*Thank you for the comments regarding our manuscript. Please see our detailed replies below.*

1. The inverse methodology description is very weak. There is no description of which search method is used! What is the combined objective function? A detailed sensitivity analysis has to be given, especially in light of the mentioned problems of equifinality. It is extremely unlikely that all 24 parameters are sensitive and justify optimization. Also inverse modelling offers the opportunity to provide the reader with an estimate of the confidence intervals for each estimated parameter, which will also reveal the sensitivity and associated uncertainty.

*With respect to the objective function, the central theme of the paper was to employ a standard publicity available model to test our hypothesis, not to devise new algorithms for inversion, and that is why we did not get into the inverse modeling details in great depth. As it was mentioned in the paper, more description about inverse modeling can be found in Mualem (1976), van Genuchten (1980), and Turkeltaub et al. (2015).*

*Moreover, Wang et al (2009) have done a detailed sensitivity analysis of groundwater recharge and evapotranspiration for soil hydraulic parameters in a single layer. The objective function we used was minimizing RMSE between observations and the model using the standard optimization algorithm provided in the HYDRUS software (L234). We provided the other goodness-of-fit metrics to further test and evaluate the model fit.*

*Thank you for the suggestions on the sensitivity analysis, it was very enlightening and has greatly improved the manuscript. In order to investigate the key sources of error, as you suggested, we performed a set of sensitivity analysis experiments of effects of soil hydraulic parameters and plant root growth on the ETa and results are presented in part 3.3 (L383-L417) and figures 10 and 11. We indeed found that 3 (alpha, n, Ks) of the 6 parameters were the most sensitive for the 4 soil layers. When preforming a full optimization with all 24 parameters we had an ETa RMSE of 0.911 mm/day compared to 1.511 mm/day using only 12 parameters (L402). Given that alpha, n, and Ks in the top layer were most sensitive, the CRNP may be beneficial to constrain these parameters during periods dominated by evaporation (L394). We also note that more in depth sensitivity analyses and multiple objective functions could be performed in the future as an extension of this work (L407).*

*Lastly, 95% confidence intervals were provided to all parameter estimates in Table 5.*

*Wang, T., V. A. Zlotnik, J. Simunek, and M. G. Schaap. 2009. Using pedotransfer functions in vadose zone models for estimating groundwater recharge in semiarid regions. Water Resources Research  45: 12. doi:10.1029/2008wr006903.*

2. The results of simulated SWC seems to be reasonable from a SWC perspective, but it's important to also address the certainty/robustness and likelihood of the estimated soil parameters. Are they random parameter picks from an equifinal problem or are they physically reasonable and do their mutual differences fit into field/lab measurements (I assume soil samples exists from the sites)? The author has attempted to validate the spatial distribution of the estimated soil parameters based on a soil map, which is highly appreciated. However, it would have been interesting to utilize this information for regionalizing the soil parameters and thereby limiting the number of free parameters in the calibration. Likewise, the soil map could have been used to upscale the AET simulations to the field scale by including the soil map instead of a simple average of the four points.

*Thank you for the suggestion. As you suggested we upscaled ETa based on the SSURGO soil map and added the results in the manuscript (L329-L331, L350, and L363), figure 9, and tables 6 and 7. We also note that a set of hydrogeophysical maps using electromagnetic and cosmic-ray neutron rovers exist for the site and will be investigated in a companion manuscript in the future. Preliminary results indicate SSURGO zone definition is fairly accurate compared to the hydrogeophysics. However, certain boundaries appear off as a result of the limited information built into the SSURGO delineation. It is unclear how far off this lines are and if the hydrogeophysics can improve this lines for applications like precision agriculture.*

3. The results of the AET simulations seem to be very poor. I miss a critical view on the results regarding lacking ability to simulate even inter-annual variability (fig 11) and perhaps more importantly the apparently complete lack of predictive capability on the daily scale. The performance metrics in Table 6 indicate good $R^2$ and NSE, but that correlation is intrinsically given by the seasonality of the climate. The real test is if the model has any predictive power on estimating the evaporative fraction AET/PET. If you normalize the AET on a daily timescale by the daily PET and then calculate the $R^2$ and NSE, you probably get no explanation of variance. This can also be somewhat illustrated in table 6, if you add a column of RMSE in % of average daily AET, then you see that the RMSE is in the order of 50-80% of the daily AET (see attached table). In comparison most Remote sensing AET methods can, with calibration, achieve results in the order of RMSE of 25-30% of the daily mean AET.

*We compared the results with EC measured ET in this study just as a simple comparison as there was no other relatively accurate measured ET data available in the study area. As it was mentioned in the paper there are always different uncertainties involve in the Eddy-Covariance (EC) measurements. EC measurements can be bias by up to 20% or*

*even more. Considering this we cannot easily say since the simulated ET values are not perfectly matched with EC ET measured data that "the AET simulations seem to be very poor". We have now provided guidance on the goodness-of-fit RMSE metrics for both SWC (L278-281) and ETa (L331-333).*

*L278-L281: "In this research we define RMSE values less than 0.03 cm³/cm³ between observed and simulated SWC values as well-matched and RMSE between 0.03 and 0.06 cm³/cm³ as fairly well-matched. We note the target error range of satellite SWC products (e.g. SMOS and SMAP) is less than 0.04 cm³/cm³ (Entekhabi et al., 2010).*

*L331-333: "In this research we consider RMSE values less than 1 mm/day between observed and simulated $ET_a$ values as well-matched and RMSE values between 1 and 1.2 as fairly well-matched (Figure 9 and Table 6)."*

*Your suggestions are appreciated and if we had access to more accurate measured ET data, like Lysimeter measured ET, we could investigate such analysis. Because of the nature of EC ET measurements (which is not based on Kc values, but instead based on the flux measurements) such comparison may not be useful. As an example obtained Kc values from EC (2007-2012) are shown below. According to the graph, most of the times during mid-growing season, obtained Kc values from EC are less than 1 (we usually expect to have values of 1-1.2 during the mid-growth season). The average EC Kc value during July and August (2007-2012) is 0.81 with a minimum average Kc value of 0.58 in 2012 and maximum Kc value of 0.99 in 2011. On the other hand, sometimes Kc values exceed 4 while in the "real world" such Kc values do not exist. In addition, Kc values do not usually change suddenly during the growing season and it is rarely possible to have a Kc value of 1 in one day and Kc value of 0.4 for the next day, but according to the graphs in some of the days we can see this case in the EC Kc values. The inherent noise seen in Kc makes this comparison challenging without temporal smoothing.*

[Figure]

4. Given the very little detail available on the AET model used (Feddes 1978) I can only speculate, but perhaps the simulated SWC is not accurate enough at the critical moments when AET is limited by water availability, or the AET model is not appropriate or the climate data are poor. But overall I do not find the results on simulated daily AET encouraging. An uncertainty analysis of the different model components would be appropriate (see comment below).

*Thank you for the suggestion. We added more details about Feddes (1978) model in the manuscript and that should make ETa estimation process clearer (L171-L176). Also, as previously mentioned we performed sensitivity analysis as you requested and results are presented in part 3.3 (L383-L417) and figures 10 and 11. This included a sensitivity analysis of the maximum dynamic rooting depth on ETa. The sensitivity to root distribution is more challenging and beyond the scope of the current paper.*

*We appreciate the reviewer's thoughts, but most of the comments are made based on the apparent difference between the EC measured ET and simulated ET. The Hydrus model is a widely used method based on a solution to the Richards Equation. The Mead Site 3 flux tower is a long standing Ameriflux tower and continues to be a part of the core network. In order to address the comments, we performed a sensitivity analysis of all 24 soil hydraulic properties building on Wang et al. (2009). A full sensitivity analysis of the root model parameters is beyond the current scope of the paper and we refer the reviewer to Guswa (2012) and Rosolem et al. 2012 for a more robust treatment.*

*L171-176: "The root water uptake, S(h), was simulated according to the model of Feddes et al. (1978)*

*$$S(h) = \alpha(h)S_p \qquad\qquad\qquad (4)$$*

*where $\alpha(h)$ is the root-water uptake water stress response function, is dimensionless and varies between 0 and 1 depending on soil matric potentials, and $S_p$ is the potential water uptake rate and assumed to be equal to $T_p$. The summation of actual soil evaporation and actual transpiration is $ET_a$."*

*Guswa, A. J. 2012. Canopy vs. Roots: Production and Destruction of Variability in Soil Moisture and Hydrologic Fluxes. Vadose Zone Journal 11:3. doi:10.2136/vzj2011.0159.*

*Rosolem, R., H. V. Gupta, W. J. Shuttleworth, X. B. Zeng, and L. G. G. de Goncalves (2012), A fully multiple-criteria implementation of the Sobol' method for parameter sensitivity analysis, J. Geophys. Res.-Atmos., 117. doi:10.1029/2011jd016355.*

Q: footprint analysis? EC footprint of 250 m radius is very large, what is the height of the EC mast?

*Thank you for your comment. We added more information about the EC height in the manuscript (L119-L123).*

*The height on the EC mast varies with crop height. According to Suyker et al. (2004):*

*"To have sufficient upwind fetch (in all directions) representative of the cropping system being studied, eddy covariance sensors were mounted at 3.0 m above the ground while the canopy was shorter than 1.0 m, and later moved to a height of 6.2 m until harvest."*

*The footprint of the tower will there change over the season, ~100 times the tower height. This is a long running Ameriflux site and the variable footprint is a part of the method and its inherent uncertainty.*

*Suyker, A. E., S. B. Verma, G. G. Burba, T. J. Arkebauer, D. T. Walters, and K. G. Hubbard. 2004. Growing season carbon dioxide exchange in irrigated and rainfed maize. Agric. For. Meteorol. 124:1-2: 1-13. doi:10.1016/j.agrformet.2004.01.011.*

*L119-L123: "At this field, sensors are mounted at 3.0 m above the ground while the canopy is shorter than 1.0 m. At canopy heights greater than 1.0 m, the sensors are then moved to a height of 6.2 m until harvest in order to have sufficient upwind fetch (in all the directions) representative of the cropping system being studied (Suyker et al., 2004)."*

5.  Please explain the reasoning behind eq. 2 and 3?

    *We explained the reason in the response and explain in the manuscript that they are one of the model inputs (L167-L170). We needed to introduce potential evaporation (Ep) and potential transpiration (Tp) values to the Hydrus model. By using Beer's law we were able to divide ETp to Ep and Tp. Based on LAI values, with equation 2 we can calculate the Ep values and then by having the Ep value we can use equation 3 to calculate the Tp values. More information can be found in Šimunek et al, (2013).*

    *Šimunek, J., Šejna, M., Saito, H., Sakai, M., van Genuchten, M.T. (2013). The HYDRUS-1D Software Package for Simulating the One-Dimensional Movement of Water,Heat, and Multiple Solutes in Variably-Saturated Media, Version 4.17.Department of Environmental Sciences, University of California Riverside, Riverside, California, USA, 307 pp.*

6.  L168: The Actual Transpiration is calculated using Feddes 1978 based on Tp and root density distribution. That must be a key component of this approach, please give more details on the application of the Feddes model.

    *Thank you for your comment. We added more details about Feddes (1978) model in the manuscript and that should make ETa estimation process clearer (L171-L176).*

    *L171-176: "The root water uptake, S(h), was simulated according to the model of Feddes et al. (1978)*
    $$S(h) = \alpha(h)S_p \qquad\qquad\qquad\qquad (4)$$
    *where $\alpha(h)$ is the root-water uptake water stress response function, is dimensionless and varies between 0 and 1 depending on soil matric potentials, and $S_p$ is the potential water uptake rate and assumed to be equal to $T_p$. The summation of actual soil evaporation and actual transpiration is $ET_a$."*

7.  L198-204: Optimized against which objective function? What was the calibration target? Which optimization algorithm (gradient based/global etc.) is used? That has to be clear up front? Also what was the result of the sensitivity analysis? Which type of sensitivity analysis, was it necessary to optimize all parameters? And why not calibrate all four layers simultaneous?

*Thank you for your questions. We added a description of the objective function to the manuscript (L231-L235) and we explained why we calibrated the two upper layers first and then we calibrated the two deeper layers (L208-L213) based on minimizing RMSE between observations and model simulations. Also, as previously mentioned we performed a sensitivity analysis with results presented in part 3.3 (L383-417) and figures 10 and 11.*

*We note that we could not optimize all the layers simultaneously because the maximum number of parameters that we can be optimized by the Hydrus-1D model is 15. We have followed the same procedure as Turkeltaub et al. (2015) and Wang et al. (2015, 2016). Since we wanted to use standard software for parameter estimation, developing a new algorithm was beyond the scope of the paper. Certainly other algorithms that can estimate many parameters exist in hydrologic modeling (c.f. Vrugt et al. 2003).*

*Vrugt, J. A., H. V. Gupta, W. Bouten, and S. Sorooshian (2003), A Shuffled Complex Evolution Metropolis algorithm for optimization and uncertainty assessment of hydrologic model parameters, Water Resources Research, 39(8). doi:10.1029/2002wr001642.*

*L231-235: "With respect to the goodness-of-fit assessment, Root Mean Square Error (RMSE) between simulated and observed SWC was chosen as the objective function to minimize in order to estimate the soil hydraulic parameters. The built in optimization procedure in Hydrus-1D was used to perform parameter estimation."*

*L208-213: "Since Hydrus-1D is limited to optimizing a maximum of 15 parameters at once and that the SWC of the lower layers changes more slowly and over a smaller range than the upper layers, the van Genuchten parameters of the upper two layers were first optimized, while the parameters of the lower two layers were fixed. Then, the optimized van Genuchten parameters of the upper two layers were kept constant, while the parameters of the lower two layers were optimized. The process was continued until there were no further improvements in the optimized hydraulic parameters or until the changes in the lowest sum of squares were less than 0.1%."*

8. L220-224: It might be obvious, but please state clearly, which observation data the performance metrics are based on.

*Thank you for your question. We added that it is based on soil water content into the manuscript (L206).*

9. L230: How are the best defined, what are the weights and how was your combined objective function defined?

*We chose the selected optimized sets of soil parameters values based on RMSE but the other metrics were included for completeness (L231-L233).*

*L231-233: "With respect to the goodness-of-fit assessment, Root Mean Square Error (RMSE) between simulated and observed SWC was chosen as the objective function to minimize in order to estimate the soil hydraulic parameters."*

10. L265-266: Of course the upper layers are better you calibrated them first and then kept them fixed while calibrating the lower layers, so they have had significantly more freedom in the optimization. Try to calibrate the lower first and then fix them and calibrate the upper, then you might get different results.

*The Hydrus-1D can just optimize up to 15 parameters simultaneously and we decided to optimize the upper 2 layers first and then the 2 lower layers. The SWC data in the 2 upper layers has more dynamics than the 2 lower layers. As previously mentioned we performed sensitivity analysis as requested and results are presented in part 3.3 (L383-417) and figures 10 and 11. The sensitivity analysis was very insightful about model behavior indicating that n and alpha in the top zone were the most sensitive to ETa (L391-399), thus creating opportunities for use of the CRNP.*

*L391-399: "We found that n and α were the most sensitive, particularly in the shallowest soil layer. This sensitivity to the shallowest soil layer provides an opportunity to use the CRNP observations, particularly in the early growing season (i.e. when evaporation dominates latent energy flux), to help constrain estimates of n and α. As the crop continues to develop additional information in deeper soil layers should be used to estimate soil hydraulic parameters or perform data assimilation. Moreover, the CRNP may be useful in helping constrain and parameterize soil hydraulic functions in simpler evaporation models used in remote sensing (c.f. Allen et al. 2007) or crop modeling (c.f. Allen et al. 1998)."*

11. L336: "the various ETa estimation techniques performed well." I disagree.

*We softened the language (L367-L371) and added specific guidelines on their performance (L331-333).*

*L367-L371: "However, we note that given the fact that EC $ET_a$ estimation can have up to 20% uncertainty (Massman and Lee, 2002, and Hollineger and Richardson, 2005), and*

*accounting for the natural spatial variability of $ET_a$ due to soil texture and root depth growth uncertainties, the various $ET_a$ estimation techniques performed fairly well."*

*L331-333: "In this research we consider RMSE values less than 1 mm/day between observed and simulated $ET_a$ values as well-matched and RMSE values between 1 and 1.2 as fairly well-matched (Figure 9 and Table 6)."*

12. L337: "In fact, it is difficult to identify which is the clear solution if any." Please rephrase.

*We rephrased the sentence (L371-L372).*

*L371-L372: "In fact, it is difficult to identify which $ET_a$ estimation method is the most accurate method."*

13. Fig 9: How come the simulated values cannot go down to 0.20-0.25 for the Cosmic ray calibration, when that is possible for the TP calibrations?

*We corrected the figure (Figure 8).*

14. Fig 11: The proposed method seems to not capture the inter-annual variability, try to plot the annual values of EC against simulated annual values in a scatterplot to see if there is any correlation on an annual basis?

*Thank you for your comment. We deleted the figure and presented the results just in table 6.*

15. Fig 13: You need to plot the daily obs vs. simulated AET in a scatterplot, the accumulated curves gives no indication of the performance of the daily model simulations! The bias of the Scatter plot will however give you the same information as the offset in accumulated values.

*Thank you for your suggestion. We changed the figure to Scatter plot (Figure 9).*

16. Table 6: Needs units.

*Thank you for your comment. We have added units to table 3, 4, and 6.*

Replies to Anonymous Reviewer #2

*Thank you for the comments regarding our manuscript. Please see our detailed replies below.*

P6, L114-119: Mention the instrument height above canopy for the EC tower. This would serve as a reference to validate your claim of the footprint size.

*Thank you for your comment. We added more information about the EC height in the manuscript (L119-L123).*

*The height on the EC mast varies with crop height. According to Suyker et al. (2004):*

*"To have sufficient upwind fetch (in all directions) representative of the cropping system being studied, eddy covariance sensors were mounted at 3.0 m above the ground while the canopy was shorter than 1.0 m, and later moved to a height of 6.2 m until harvest."*

*The footprint of the tower will there change over the season, ~100 times the tower height. This is a long running Ameriflux site and the variable footprint is a part of the method and its inherent uncertainty.*

*Suyker, A. E., S. B. Verma, G. G. Burba, T. J. Arkebauer, D. T. Walters, and K. G. Hubbard. 2004. Growing season carbon dioxide exchange in irrigated and rainfed maize. Agric. For. Meteorol. 124:1-2: 1-13. doi:10.1016/j.agrformet.2004.01.011.*

*L119-L123: "At this field, sensors are mounted at 3.0 m above the ground while the canopy is shorter than 1.0 m. At canopy heights greater than 1.0 m, the sensors are then moved to a height of 6.2 m until harvest in order to have sufficient upwind fetch (in all the directions) representative of the cropping system being studied (Suyker et al., 2004)."*

P7, L138-139: The reference to integration of CRNP data into the NOAH LSM seems extraneous here, and would be better deleted.

*We deleted the reference from the manuscript.*

P7, L141-142: No numbers are given for the footprint size of the EC tower. So there's no way for the reader to decide if this assumption is valid or not. Further, with the assumption made, a discussion on the implications of this assumption later in the manuscript would be a good addition.

*Thank you for your comment. We added more information about the EC height in the manuscript (L119-L123). The assumption is that both the EC and CRNP are representative of the average conditions of the crop. This is by equivalency the same as to what an LSM grid would assume.*

*The height on the EC mast varies with crop height. According to Suyker et al. (2004):*

*"To have sufficient upwind fetch (in all directions) representative of the cropping system being studied, eddy covariance sensors were mounted at 3.0 m above the ground while the canopy was shorter than 1.0 m, and later moved to a height of 6.2 m until harvest."*

*The footprint of the tower will there change over the season, ~100 times the tower height. This is a long running Ameriflux site and the variable footprint is a part of the method and its inherent uncertainty.*

*Suyker, A. E., S. B. Verma, G. G. Burba, T. J. Arkebauer, D. T. Walters, and K. G. Hubbard. 2004. Growing season carbon dioxide exchange in irrigated and rainfed maize. Agric. For. Meteorol. 124:1-2: 1-13. doi:10.1016/j.agrformet.2004.01.011.*

*L119-L123: "At this field, sensors are mounted at 3.0 m above the ground while the canopy is shorter than 1.0 m. At canopy heights greater than 1.0 m, the sensors are then moved to a height of 6.2 m until harvest in order to have sufficient upwind fetch (in all the directions) representative of the cropping system being studied (Suyker et al., 2004)."*

P8, L163: Please provide references to the Beer's law.

*We added the reference to the Beer's law (L167) for the Hydrus code.*

P8, L167: It may be better to mention that the LAI was described in the previous or study area section, rather than "above".

*We changed it as you suggested (L171).*

*L171: "where k is an extinction coefficient with a value set to 0.5 (Wang et al., 2009b) and LAI ($L^2/L^2$) is leaf area index described in the previous section."*

P8, L168: A brief description of how the Feddes model makes use of the potential transpiration and the root density distribution is necessary. Further, no details of the root density used in the study are given, which should be rectified.

*Thank you for the suggestion. We added more details about Feddes (1978) model in the manuscript that should make the ETa estimation process more clear (L171-L176). Also, more information about root distribution model was provided in the manuscript (L189-L190).*

*L171-L176: "The root water uptake, S(h), was simulated according to the model of Feddes et al. (1978)*

$S(h) = \alpha(h)S_p$                                                        *(4)*

*where α(h) is the root-water uptake water stress response function, is dimensionless and varies between 0 and 1 depending on soil matric potentials, and $S_p$ is the potential water uptake rate and assumed to be equal to $T_p$.  The summation of actual soil evaporation and actual transpiration is $ET_a$."*

*L189-L190: "Finally, the Hoffman and van Genuchten (1983) model was used to calculate root distribution. Further details about the model can be found in Šimunek et al., 2013."*

P10, L199-205: What were the objective functions and methodology used to optimize these parameters? No description of any sort is provided, which makes it very difficult to assess the applicability.

*We added objective function to the manuscript (L231-L235) and we explained that why we calibrated two upper layers first and then we calibrated the two deeper layers (L207-L214). Note, we could not optimize all the layers simultaneously because the maximum number of parameters that we can be optimized by the Hydrus-1D model is 15. We have followed the same procedure as Turkeltaub et al. (2015) and Wang et al. (2015, 2016). We used RMSE as our objective function.  We performed sensitivity analysis as you requested and results are presented in part 3.3 (see L383-L415) and figures 10 and 11.*

*L231-235: "With respect to the goodness-of-fit assessment, Root Mean Square Error (RMSE) between simulated and observed SWC was chosen as the objective function to minimize in order to estimate the soil hydraulic parameters. The built in optimization procedure in Hydrus-1D was used to perform parameter estimation."*

*L207-214: "In order to efficiently optimize the parameters, we used the method outlined in Turkeltaub et al. (2015). Since Hydrus-1D is limited to optimizing a maximum of 15 parameters at once and that the ] SWC of the lower layers changes more slowly and over a smaller range than the upper layers, the van Genuchten parameters of the upper two layers were first optimized, while the parameters of the lower two layers were fixed. Then, the optimized van Genuchten parameters of the upper two layers were kept constant, while the parameters of the lower two layers were optimized. The process was continued until there were no further improvements in the optimized hydraulic parameters or until the changes in the lowest sum of squares were less than 0.1%."*

P11, L223: R-squared has a name. It is called the Coefficient of Determination. Also, while the other metrics are described in equations, R-squared is not.

*We added the name to the manuscript (L236) and we described the equation (eq 10, L240).*

P12, L230: What about R-squared?

*We changed the sentence.*

P12, L236: This may be a matter of semantics, but I feel that the subsection is better titled as "Vadoze Zone Inverse Modeling Results". You are performing inverse modeling of the vadose zone, not modeling of the inverse vadose zone.

*Thank you for the suggestion. We changed the title as you suggested.*

P12, L238/239/250: Figures 4 and 7 are interchanged. Fig. 4 shows the annual precipitation, and fig 7 shows the temporal evolution of daily SWC.

*We corrected the figure numbers (Now Figures 4 and 6).*

P12, L239: Not so clear. It may be good to mention that the large standard deviation values show this. Also, I was surprised to see that the upper layers had smaller SD values than the deeper layers! As the authors themselves mention elsewhere, the soil moisture variability is expected to reduce with depth. Any discussion on this phenomenon would be welcome.

*We modified the sentence (L254-256).*

*L254-256: "Based on the large standard deviation values (Figure 4), despite the relatively small spatial scale (~65 ha) and uniform cropping at the study site, SWC varies considerably across the site (c.f. standard deviation in Figure 4), particularly during the growing season."*

P13, L272-273: Based on the numbers in Table 3, I am not sure the data are "fairly well matched". R-squared < 0.1 in the validation period (and < 0.4) in the calibration period), along with a negative NSE, tells me that the model and observation were not behaving alike. Maybe addition of distribution-level metrics could help bring out the relationship (if any) between the two better.
Also, here, and through the rest of the discussion, the authors use terms such as "fairly well matched" or "performed well" or similar language. These are highly subjective terms, and no analyses of numbers are provided to support these statements. It is necessary to establish at the beginning of the section what the authors consider as a "good" or "fairly good" etc., performance means in terms of absolute numbers. While the performance metrics are provided in the tables, no discussion is made regarding them and the reasoning for considering a particular statistic good.

*Thank you for your comments. We defined each error term for SWC and ETa, and added a section at the beginning to explain those error terms (L278-281 and L331-333).*

*L278-281: "In this research we define RMSE values less than 0.03 $cm^3/cm^3$ between observed and simulated SWC values as well-matched and RMSE between 0.03 and 0.06 $cm^3/cm^3$ as fairly well-matched. We note the target error range of satellite SWC products (e.g. SMOS and SMAP) is less than 0.04 $cm^3/cm^3$ (Entekhabi et al., 2010)."*

*L331-333: "In this research we consider RMSE values less than 1 mm/day between observed and simulated $ET_a$ values as well-matched and RMSE values between 1 and 1.2 as fairly well-matched (Figure 9 and Table 6)."*

P14, L282: How do these soil hydraulic parameters obtained from the inverse estimation compare with the textures used in the optimization? Further, while you mention earlier in the text that 6 different soil textures were used in the optimization, you omit mentioning which textures they are.

*The six textures are now included in the manuscript (L217-218). The difference in the optimized hydraulic properties roughly match with the SSURGO textural descriptions (comparison of Table 1 vs. 5 and see discussion in L299-321).*

*L217-218: "including sandy clay loam, silty clay loam, loam, silt loam, silt, and clay loam"*

P14, L289: Provide a reference or hyperlink to the Web Soil Survey Data.

*We added the hyperlink to the text (L107).*

P15, L315: The infiltration rate in fine textured soil is lower, leading to higher surface runoff, as the authors mention. However, the water holding capacity of such soils is higher than coarse soils, leading to higher stored volume. I think a better argument here may be that the plant/root would have to overcome higher pressures to extract water from the fine soil, thus leading to lower ET.

*Thank you for your suggestion. We added that to the manuscript (L338-L341).*

*L338-341: "Here smaller $ET_a$ rates at TP 4 location are likely due to finer soil texture at this location which makes it more difficult for the plant/roots to overcome potentials to extract water from the soil, thus leading to a lower $ET_a$ rate and greater plant stress."*

P16, L330: Do you mean Figures 11 and 12 here? Figure 11 is never discussed in the entire manuscript.

*We changed the figure numbers.*

P16, L330-334: generally, the phenomenon of roots extracting water from deeper layers is seen in more mature vegetation such as trees, and not in seasonal agricultural crops. Also, even accepting that the plants may be drawing from layers deeper than the model domain, the phenomenon should not be so apparent in the clayey soils (TP4). A clayey soil restricts root penetration, and usually a shallow root depth is seen in such soils.

*Thank you for your comments. We explained the phenomenon in more detail in the manuscript (L363-L367) and also we performed a root growth sensitivity analysis and presented the results in part 3.3 (L408-L417) and Figure 11. We also refer the reader to Guswa (2012) for a more complete discussion.*

*L363-367: "This shows that although 2012 was a very dry year, the plants found most of the needed water by extracting water from deeper soil reservoirs. As previously mentioned we defined a maximum root depth for the model that could greatly impact the results. To further illustrate this point, a sensitivity analysis was performed on the maximum rooting depth and presented in the following section."*

*L408-417: "A sensitivity analysis of $ET_a$ by varying rooting depth is illustrated in Figure 11. As would be expected with increasing rooting depth, higher $ET_a$ occurred. In addition, Figure 11 illustrates a decreasing RMSE against EC observations for up to 200% increases. Again it is unclear if the EC observations are biased high or in fact rooting depths are much greater than typically considered in these models. The high observed EC values in the drought year of 2012 indicate that roots likely uptake water from below the 1 m observations. Certainly the results showed here further indicate the importance of root water uptake parameters in VZMs and LSMs, even in homogeneous annual cropping systems. While beyond the scope of this paper we refer the reader to the growing literature on the importance of root water uptake parameters on hydrologic fluxes (c.f. Schymanski et al. 2008 and Guswa 2012)."*

*Guswa, A. J. 2012. Canopy vs. Roots: Production and Destruction of Variability in Soil Moisture and Hydrologic Fluxes. Vadose Zone Journal 11:3. doi:10.2136/vzj2011.0159.*

P16, L337: Clear solution to what?

*We soften the language (L367-L371).*

*L367-L371: " However, we note that given the fact that EC $ET_a$ estimation can have up to 20% uncertainty (Massman and Lee, 2002, and Hollineger and Richardson, 2005), and accounting for the natural spatial variability of $ET_a$ due to soil texture and root depth growth uncertainties, the various $ET_a$ estimation techniques performed fairly well."*

Figures and Tables: I feel that, overall, the number of figures and tables can be reduced. As mentioned earlier, Figures 4 and 7 are interchanged.

*Thank you for your comments. We tried to reduce the figures and tables and deleted some of the figures. Since we performed a sensitivity analysis we added additional figures per the reviewers suggestions.*

Figures 5 and 6: Keep any one of these two. No extra information is extracted by having two figures showing the same information here.

*We kept just figure 5.*

Figure 10: This can be merged with fig. 1.

*We merged with Figure 1.*

Figure 11: This figure is never discussed in the text. Figure 12: This could be merged into fig. 11 as another panel. Also, in the text, this figure is discussed after fig. 13.

*We changed the figures and discussion as suggested.*

Table 1: These numbers can be discussed in the text instead of adding a single row table. As mentioned in an earlier comment, almost none of the numbers from the tables are discussed in context.

*Thank you for the suggestion. We have decided to keep table to make it easier to understand expected range of parameters and understanding of sensitivity analysis presented.*

Table 3: Can be merged with tab. 2.

*Thank you for suggestion. Since calibration and validation are different years we decided to keep both tables for clarity.*

Table 7: There is no need for this table. The numbers can be mentioned in figs. 11 and 12. That would also make those figures easier to interpret.

*Thank you for suggestion. We decided to keep the tables for clarity instead of including numbers in text and in other tables.*

Technical comments:
P4, L75: Should read as "… hyper-resolution LSM grid cells…" P5, L93: Check the spelling of the name "Simunek". P7, L135: "The CRNP measurement depth…" P7, L147: "… explained in detail by …" P9, L176: "… GDD approximately 60-70…" P10, L195: The abbreviation TP has not been established earlier. P10, L198: the parameter "l" should be in lower case. P13, L262: "… criteria at TP locations…" P15, L302: "… inverse VZM modeling…" VZM already includes model. P16, L333: "VZM model" Same as above. References: Ensure uniform formatting of all the bibliography. Some end in page numbers, some in years, some in journal names, and some in volumes/ issues. Table 4, Column 8: Use lower case "l" for tortuosity. Table 5, Column 6: Hectares in Field.

*We have made the changes, thank you.*

Replies to Anonymous Reviewer #3

*Thank you for the comments regarding our manuscript. Please see our detailed replies below.*

Authors estimated field scale evapotranspiration (ET) by calibrating a 1D unsaturated zone model (HYDRUS-1D) using soil water content measurements, and compared simulated ET with observed ET from an eddy covariance tower. The HYDRUS-1D soil hydraulic parameters were calibrated using daily soil water content measurements from four theta monitoring probes at multiple depths and one cosmic ray neutron probe. While this is an interesting study, the novelty of the current study is not clear. Based on presented results, large differences exist between simulated ET and eddy covariance data and results of soil moisture simulations are not entirely satisfactory given the negative NSE during calibration and small coefficient of determination for soil moisture simulations at certain depths. In particular, authors have not discussed the implications of their results and what can be done to improve model estimation. While the focus of the inverse modelling was on soil hydraulic parameters estimation, the study can benefit from a detailed model sensitivity experiment to soil hydraulic and root growth function model parameters. I suggest authors to perform a detailed uncertainty estimation approach to identify the sources of errors (model input, parameters, or model structure) in

ET and soil water content estimates. This can help to identify why the model did not perform well in some cases and how authors can improve their results.

*Thank you for the suggestions. In order to investigate the key sources of error, as you suggested, we performed a set of sensitivity analysis experiments of effects of soil hydraulic parameters and plant root growth on the ETa and results are presented in part 3.3 (L383-L417) and figures 10 and 11. The preliminary sensitivity analysis on a number of parameters was very insightful and has improved the manuscript considerably. We also provide a description and a few key citations (Bastidas et al. 1999 and Rosolem et al. 2012) to undertake a more in depth sensitivity analysis in future work.*

1. Introduction, the rational and implications of the current study are not entirely clear. I suggest authors outline the main objectives of their study and discuss how their results advance our understanding of ET estimation using unsaturated zone models. It is not clear whether authors try to develop a benchmark for soil moisture or ET estimation or how their soil hydraulic parameter estimation can help parametrize hyper-resolution land surface models? These are the ideas that are discussed in the Introduction but their links with the current study are not clear.

*The introduction was modified for clarity. Specifically, the manuscript lays out the methodology for taking SWC observations to estimate ETa. Both SWC and ETa are key benchmarks needed by the LSM community. The final paragraph in the introduction lays out the motivation and key outcomes (L89-100). The abstract also discusses the societal need for these value added products from SWC monitoring alone.*

*L89-100: "The aim of this study is to examine the feasibility of using inverse VZM modeling for estimating field scale $ET_a$ based on long-term local meteorological and SWC observations for an Ameriflux (Baldocchi et al., 2001) EC site in eastern Nebraska, USA. We note that while this study focused on one particular study site in eastern Nebraska, the methodology can be easily adapted to a variety of SWC monitoring networks across the globe (Xia et al., 2015), thus providing an extensive set of benchmark data for use in LSMs. The remainder of the paper is organized as follows. In the methods section we will describe the widely used VZM, Hydrus-1D (Šimunek et al., 2013), used to obtain soil hydraulic parameters. We will assess the feasibility of using both profiles of in-situ SWC probes as well as the area-average SWC technique from Cosmic-Ray Neutron Probes (CRNP). In the results section we will compare simulated $ET_a$ resulted from calibrated VZM with independent $ET_a$ estimates provided by EC observations. Finally a sensitivity analysis of key soil and plant parameters will be presented."*

2. Section 2.2.1. It seems authors have used a different growth root model compared to the HYDRUS-1D root growth model for annual vegetation. Have authors performed any experiments to assess how the results of the two root growth models compare?

   *Since we had annual cultivation rotation between soybean and maize we had to introduce the root depth to the model and we could not use the default values inside the model. Likewise, as default values were constant and cannot be changed for different type of crops in different years during the simulation, we were not able to compare the models. This parameterization is not available in the standard HYDRUS package and a limitation of using it with crop rotations. We wanted to keep intact the cropping history to minimize impact on SWC between years. Clearly the topic of root water uptake deserves more investigation. We did perform a root depth sensitivity analysis summarized in Figure 11.*

3. Section 2.2.1. It will be very useful if authors can report Kc parameters and root growth model parameters as they can impact the results of ET estimation.

   *As it was mentioned, we performed a root growth sensitivity analysis and presented the results in part 3.3 (L408-L417) and Figure 11 to investigate the impact of root depth on ETa. A discussion of observed Kc values can be found above in response to reviewer 1 (graph on page 6).*

4. Section 2.2.2. Additional details regarding the inverse modelling algorithm and an objective function that is used for parameter estimation are required.

   *We added a description of the objective function to the manuscript (L233-L235) and we explained why we calibrated the two upper layers first and then we calibrated the two deeper layers (L209-L212) based on minimizing RMSE between observations and model simulations. Also, as previously mentioned we performed a sensitivity analysis with results presented in part 3.3 (L383-417) and figures 10 and 11.*

   *We note that we could not optimize all the layers simultaneously because the maximum number of parameters that we can be optimized by the Hydrus-1D model is 15. We have followed the same procedure as Turkeltaub et al. (2015) and Wang et al. (2015, 2016). Since we wanted to use standard software for parameter estimation, develop a new algorithm was beyond the scope of the paper. Certainly other algorithms that can estimate many parameters exist in hydrologic modeling (c.f. Vrugt et al. 2003).*

   *Vrugt, J. A., H. V. Gupta, W. Bouten, and S. Sorooshian (2003), A Shuffled Complex Evolution Metropolis algorithm for optimization and uncertainty assessment of*

*hydrologic model parameters, Water Resources Research, 39(8). doi:10.1029/2002wr001642.*

5. Section 2.2.2. Line 206- Can authors provide further details about initial soil hydraulic parameters that they used in the modelling experiment? Did they use soil hydraulic parameters based on soil texture class information? Similarly, authors used the same parameter bounds for model calibration for all soil texture classes. It will be useful if authors can incorporate the soil texture information to define priors and initial parameter values.

*The initial values were just the default values in the Hydrus-1D model which are based on the different soil types. Agreed, priors could be used with pedotransfer functions to improve results. Unfortunately, the connection between hydrologic fluxes and soil texture classes is unclear (Groenendyk et al. 2015). This work continues on that disconnection. The comparison between the SSURGO textural classes and the optimized soil hydraulic functions (Tables 1 to 5) deserves more attention in future work.*

*Groenendyk, D. G., T. P. A. Ferre, K. R. Thorp, and A. K. Rice. 2015. Hydrologic-Process-Based Soil Texture Classifications for Improved Visualization of Landscape Function. PLoS One 10:6: 17. doi:10.1371/journal.pone.0131299.*

6. Section 2.2.2. Why homogeneous soil type was used for simulating water content for the Cosmos-Ray neutron probe while for the Theta probes variability in vertical hydraulic conductivity is considered?

*As a first cut we used a single layer. Since the CRNP only sees the top 20 cm we wanted to see how well it could or not reproduce ETa values. Clearly more investigation is needed about the use of CRNP to estimate ETa. The sensitivity analysis indicates that constraining alpha, n, and Ks in the top layer is most important for estimates of ET. The CRNP could be used to estimate these in periods where evaporation controls Latent Energy flux as suggested in L391-393.*

*L391-393: "Moreover, the CRNP may be useful in helping constrain and parameterize soil hydraulic functions in simpler evaporation models used in remote sensing (c.f. Allen et al. 2007) or crop modeling (c.f. Allen et al. 1998)."*

7. Why the spin-up period is varied between the inverse modelling approach and the forward model? What criteria authors used to define model spin-up?

*Because the longer sets of climatic data exist, compared to the SWC at the study site (L247-L247).*

*L247-L249: "Finally, we note that the years 2004-2006 were used as a model spin-up period for the forward model and evaluation of ET$_a$ because of the longer climate record length."*

8. Table 2-Why negative NSE is obtained during calibration period particularly in deeper soil layers? Even R2 values are pretty small for a VZM model that is calibrated to observations. Can authors describe the reasons for this mismatch? Similarly results of soil moisture simulation are not satisfactory for the CRNP calibration based on Table 3.

*We defined each error term for SWC and ETa, and added a section at the beginning to explain those error terms (L278-281 and L331-333). We note that the model is optimized by RMSE whereas NSE and R2 are additional evaluation metrics. We deemed well matched as RMSE between 0 and 0.03 cm3/cm3 per satellite remote sensing standards. With respect to the difference in ETa, I suspect the root zone depth and distribution will greatly impact this as indicated by our preliminary sensitivity analysis (Figure 11). Clearly more work devoted to root water uptake parameters is needed.*

*L278-281: "In this research we define RMSE values less than 0.03 cm$^3$/cm$^3$ between observed and simulated SWC values as well-matched and RMSE between 0.03 and 0.06 cm$^3$/cm$^3$ as fairly well-matched. We note the target error range of satellite SWC products (e.g. SMOS and SMAP) is less than 0.04 cm$^3$/cm$^3$ (Entekhabi et al., 2010)."*

*L331-333: "In this research we consider RMSE values less than 1 mm/day between observed and simulated ET$_a$ values as well-matched and RMSE values between 1 and 1.2 as fairly well-matched (Figure 9 and Table 6)."*

9. Authors indicate that inverse modelling based on CRNP data is most useful during the periods that soil evaporation is dominant. Can authors further explain why that is the case? One would expect that CRNP should provide better estimate of ET as its footprint is likely to overlap the EC tower footprint.

*Since the CRNP only sees the top 20 cm we wanted to see how well it could or not reproduce ETa values. We hypothesize that at roots development into deeper layers and Transpiration becomes more important in the latent energy term the information content in the CRNP would diminish. Clearly this topic requires more investigation. The sensitivity analysis indicates that constraining alpha, n, and Ks in the top layer is most*

*important for estimates of ET. The CRNP could be used to estimate these in periods where evaporation controls Latent Energy flux as suggested in L397-399.*

*L397-399: "Moreover, the CRNP may be useful in helping constrain and parameterize soil hydraulic functions in simpler evaporation models used in remote sensing (c.f. Allen et al. 2007) or crop modeling (c.f. Allen et al. 1998)."*

10. Section 3.2. Authors relate variability in performance of the model in ET simulation to variability in soil texture. However, one important information that is missing is vegetation type at the location of the probes and the EC tower footprint scale. Perhaps, authors should combine ET estimates from multiple probes to estimate ET at a field scale.

    *As you suggested we upscaled ETa based on the SSURGO soil map to the field scale and results added in the manuscript (L329-L331, L350, and L363), Figure 9, and tables 6 and 7.*

11. It will be useful if authors can provide information about deep drainage from model simulations at multiple locations.

    *For the interested reader, the deep drainage can be calculated by the using mass balance with precipitation, ET, and runoff provided in the manuscript (L341-345). Since deep drainage was not discussed in the results it is unclear what this would provide to the main objective of the paper. For detailed discussion of deep drainage in Neb we suggest the reader see Wang et al. 2016.*

    *L341-345: "In addition, higher surface runoff can be expected at the TP 4 location due to finer-textured soils. According to the simulation results the average surface runoff at the TP 4 location was about 44.8 mm/year from 2007 to 2012, while the average surface runoff at the other three locations (TPs 1-3) was around 10.6 mm/year, which partially accounts for the lower $ET_a$ rates."*

    *Wang, T., Franz, T. E., Yue, W., Szilagyi, J., Zlotnik, V. A., You, J., et al. (2016). Feasibility analysis of using inverse modeling for estimating natural groundwater recharge from a large-scale soil moisture monitoring network. Journal of Hydrology, 533, 250-265.*

12. Line 166- Extinction

*Change made, thank you.*

13. Line 238-Please revise the Figure number to 7.

    *We corrected the figure numbers (Figures 4 and 6).*

14. Figure 5- Can authors describe the reason for large differences between the spatially averaged TP and CRNP by the end of year 2014?

    *We suspect that there is an issue with the TP data at that point in time, perhaps due to frozen soils?*

[revised manuscript text omitted]
 \\[2em] or\ D = MRDD = \dfrac{AGDD}{GDD_{Silking}}MRD \\[2em] \qquad\qquad else \\[2em] \qquad\quad D = MRD \end{cases}$$

**Page 18: [2] Formatted**      Trenton Franz      11/28/16 2:41 PM

Font:Italic

**Page 18: [3] Formatted**      Trenton Franz      11/28/16 2:41 PM

Font:Italic, Subscript

**Page 18: [4] Formatted**      Trenton Franz      11/28/16 2:41 PM

Font:Italic

**Page 18: [5] Deleted**      Trenton Franz      11/21/16 11:19 AM

which were performed to identify the sources of the errors in both $ET_a$ estimation and soil hydraulic parameters optimization processes

**Page 18: [6] Formatted**      Tiejun Wang      12/1/16 4:17 PM

Subscript

**Page 18: [7] Formatted**      Tiejun Wang      12/1/16 4:17 PM

Subscript

**Page 18: [8] Deleted**      Trenton Franz      11/21/16 11:24 AM

to identify the sources of the errors in $ETa$ estimation and

**Page 18: [9] Deleted**      Trenton Franz      11/21/16 11:27 AM

Based on the analysis for homogeneous single soil layer and 4-layer soil profile (F

| Page 18: [10] Formatted | Trenton Franz | 11/28/16 2:41 PM |
|---|---|---|

Font:Italic

| Page 18: [11] Formatted | Trenton Franz | 11/28/16 2:41 PM |
|---|---|---|

Font:Italic, Subscript

| Page 18: [12] Deleted | Trenton Franz | 11/21/16 11:28 AM |
|---|---|---|

s on both *ETa* estimation and soil hydraulic parameters optimization except

| Page 18: [13] Deleted | Trenton Franz | 11/21/16 11:28 AM |
|---|---|---|

which did not have a significant effect on the results

| Page 19: [14] Deleted | Trenton Franz | 11/21/16 4:59 PM |
|---|---|---|

Even though we discovered that almost all the soil hydraulic parameters have effects on *ETa* estimation, we tried to limit the optimized factors (e.g., we just optimized $\alpha$, *n, K$_{s,}$* and used model default for the other parameters) to see if that can help us to improve the results and obtain more accurate *ETa* estimation but after simulations we acquired higher RMSE values between measured and simulated *ETa* values. In fact, the results showed that more model input would help to have more robust output which are soil hydraulic parameters and *ETa* values here in this research.

Also, effect of root depth growth was investigated to see if it has an effect on simulations. Results of

| Page 48: [15] Deleted | Trenton Franz | 11/29/16 10:51 AM |
|---|---|---|

---

## Author Response (AR2)

Dear Prof. McCabe,

We would like to thank you and the two reviewers for their time and excellent comments regarding our manuscript, titled "Feasibility analysis of using inverse modeling for estimating field-scale evapotranspiration in maize and soybean fields from soil water content monitoring networks". After careful analysis of all the comments, we have made revisions to our manuscript and added a section better explaining the application and limitations associated with the method (3.4). You can find our detailed responses to the reviewers' comments (shown in red italics) and the changes we made to the manuscript in the following sections. We have also included a marked up version of the original manuscript.

On the behalf of all coauthors, I hope that this revised version would meet the publication standard of Hydrology and Earth System Sciences (HESS) and inclusion in the Eric F. Wood special issue. Please let us know if there are more questions and comments about the manuscript.

Sincerely,

Prof. Trenton E. Franz
School of Natural Resources
University of Nebraska-Lincoln, USA

Reply to the editor:

*Thank you for the comments regarding our manuscript. Please see our detailed replies below.*

As you will see, I have received two referee report on your revised manuscript. Both appreciate the significant effort that has gone into improving this resubmitted version.

Referee #1 has reiterated a request for some further interpretation of results, ideally in a separate Discussion section. I tend to agree that providing some additional insight into the issues related to your approach, as well as any limitations or advantages, would be useful to the reader. Whether this should be done in a separate Discussion section I will leave to you, but it would be good to see some of your thoughts on the challenges (and opportunities) that may be presented by this approach, as well as addressing the issues raised by Ref #2.

*Thank you for the suggestions. We added a separate section (3.4) and explained the applications and limitations of the method in more details.*

Referee #2 is largely happy with the changes, but identifies a number of relatively minor corrections in their report. These should be fairly straightforward to address.

*Thank you for the comments. We have made the changes as requested.*

Overall, I believe the requested changes will not present any major challenges to implement and expect that I will be able to make a final Editorial decision upon receipt of an updated manuscript and a detailed response to the reviewers.

*We really appreciate you taking time and reviewing our manuscript. We look forward to hearing back from you about a final decision.*

Replies to Anonymous Reviewer #2

*Thank you for the comments regarding our manuscript. Please see our detailed replies below.*

This is a revised manuscript, and the authors have put in significant effort to address the comments raised in the earlier round of review. As such, the manuscript appears to be vastly improved in terms of comprehensiveness and flow. However, I do have a few minor points that I feel will further improve the manuscript if addressed. I list them below. Apart from this, a further scan for grammar and language would be beneficial. My recommendation to the editor is to accept the manuscript for publication in HESS subject to the following minor concerns being addressed by the authors.

*Thank you for your time and comments. We have modified some of the sentences based on your comments and added a few more details on $ET_r$ calculations for clarity.*

Abstract, L27: What is the "plumber experiment"? This is the only location it appears in the manuscript. There is no mention of this experiment in the main text or in the references. If you are mentioning something in the abstract, I would expect it to be materially significant to the narrative, and to read more about it in the introduction section (in this case).

*Thank you for the comment. You are right, we never used the term in the manuscript so we just decided to edit the sentence and avoid having the term "plumber experiment" within the abstract.*

P5, L89: "… VZM modeling…" Does the M in VZM not stand for model/ing?

*Yes, we have deleted the repeated term (L93).*

P7, L133: Why was the ETr computed for alfalfa, and not for either maize (a tall crop itself) or soybean? No justification is provided for using an ETr from a different crop to the one grown in the particular field. Even if it is a standard procedure to use alfalfa for ETr, and then use crop coefficients to translate the ETr for the particular crops, it is necessary to mention that here.

*Thank you for your comment. As you mentioned, based on ASCE Penman-Monteith equation (which is a standard procedure), we can just compute reference ET ($ET_r$) for either grass or alfalfa and then using crop coefficient ($K_c$) values we can calculate $ET_p$ values for each individual crop. Here we computed $ET_r$ values for alfalfa and then using crop coefficient ($K_c$) values suggested by Allen et al. (1998) and Min et al. (2015) we calculated $ET_p$ values for maize and soybean. We have tried to make it clearer in the manuscript that we have indeed followed a standard procedure here (L167-169).*

*L167-169: "Based on ASCE Penman-Monteith equation, $ET_r$ values can be computed for either grass or alfalfa and then using crop-specific coefficients daily potential evapotranspiration ($ET_p$) can be calculated."*

P7, L143: You mention that the CRNP measurement depth varied between 15 and 40cm, but assume a mean depth of 10cm over the entire period. Why was the mean depth assumed to be outside the 15-40cm range?

*Thank you for your comment and sorry for the confusion. The depth varied between 15-40 cm but as we mentioned in the manuscript for simplicity purposes we assumed the CRNP has an effective depth of 20 cm meaning the CRNP soil water content measurement comes from the top 20 cm soil layer. This is a conservative guess for the depth of the sensor. Moreover, we also assign a node at 10 cm for the inverse modeling and estimate of hydraulic properties from a 0 to 20 cm soil depth. This node depth must be assigned in the Hydrus modeling framework.*

P11, L225: Contrary to the above point, you mention that the average effective measurement depth was considered to be 20cm, and an observation point was set at 10cm depth. This is confusing.

*Please see comment above.*

P18, L372: I am happy you bring up the point about equifinality, and thus imply that deterministic modeling may not always be the best option in hydrology. Thank you for discussing this possibility at this juncture.

*Thank you for the comment.*

Replies to Anonymous Reviewer #3

*Thank you for the comments regarding our manuscript. Please see our detailed replies below.*

Authors have addressed editor and reviewers' comments in the revised manuscript. Performing sensitivity experiments provided further insights about the modeling results. However, two main issues still remain.

1) While authors report other statistical measures for model performance, they only focused on RMSE values for discussing their results. As shown in Table 3 and raised by reviewer 3 in the first set of comments, it is problematic to have negative NSE values. This means that average observed values are better than model simulation. Authors need to discuss this disagreement.

*Thank you for your comment. As it is mentioned in the manuscript RMSE was chosen as the objective function and MAE, NSE, and $R^2$ are used as the additional evaluation metrics. However, you are right that negative NSE values mean that the simulation results are not as good as mean observed values. We tried to improve NSE values by minimizing the inaccuracy but there are many free parameters involved in modeling that can affect the results. Since we selected RMSE as our objective function (very often done in literature) we felt additional metrics were useful for the reader to gauge the level of goodness-of-fit. If NSE would have been selected*

*as our objective function you would likely get a different set of parameters. We considered a multi-objective optimization procedure using a compromised solution but thought it was beyond the scope of the current paper. Clearly a weakness in any multiple objective optimization is selection of the objective function(s).*

2) As it was raised by the editor and reviewers, a detailed discussion section regarding the limitation and advantages of this modeling study is required. In particular, how combination of CRNP and point measurements of soil moisture observations can be used to provide field scale estimates of Eta.

*Thank you for your comment. We added a separate section (3.4) (L427-454).*

*L427-454: "**3.4 Applications and limitations of the vadose zone modeling framework***

*Given its simplicity and widespread availability of ground data, $ET_r$ and Kc values are often used in a wide variety of applications to estimate $ET_p$ and thus approximate $ET_a$. It is well known that SWC is a limiting factor affecting the assumption that $ET_p \sim ET_a$. On the other hand, we know that SWC observations are local in nature and not necessarily representative of $ET_a$ footprint estimates. The key questions are: what is the value of SWC observations, how many profiles do we need to install in a footprint, and at which depths to constrain estimates of fluxes? The well instrumented and long-term study presented here allows us to start to answer these key questions. First we find that $ET_p$ has an average annual value of 1064.9 mm as compared to EC at 612.5 mm (Table 7). By including individual SWC profiles (TP 1 to 4) and the CRNP in the VZM framework we are able to constrain our estimate of $ET_a$ to between 525.3 and 643.1 mm and reduce $ET_a$ RMSE from 1.992 mm/day to around 1 mm/day (Table 6). In addition, a range of soil hydraulic parameters for each depth and spatially averaged top layer can be estimated to help better constrain recharge fluxes simultaneously. Given the principle of equifinality in hydrologic systems, the VZM framework may lead to equally reasonable estimates of parameters which is a limitation of the method and LSMs in general. Based on our sensitivity analysis (Figure 10) the key parameters of α, n may greatly affect $ET_a$.*

*Although sparsely distributed, widespread state, national, and global meteorological observations paired with SWC profiles (Xia et al. 2015) and the VZM framework provide an opportunity to better constrain $ET_a$ and local soil hydraulic functions. Moreover, where multiple SWC profile information is available a range of $ET_a$ and soil hydraulic parameters can be estimated and thus considered in LSM data assimilation frameworks. The combination of basic metrological observations with a CRNP in the VZM framework further allows for estimates of upscaled soil hydraulic parameters with similar estimates of $ET_a$ as found with individual SWC profiles. Moving forward, combining CRNP with deeper SWC observations from point sensors seems to be a reasonable strategy in order to average the inherent SWC variability in the near surface yet provide SWC constraints at depth, particularly as annual crops develop over the growing season."*

In Figure 4- Why standard deviations of water content in deeper soil layers are higher than shallow soil layers? As reviewer 1, suggested it would be worthwhile to try calibrating deeper soil layers first and then focus on shallow soil layers to assess model performance.

*Thank you for your comment and sorry for the confusion. In Figure 4 we tried to show the temporal SWC variability throughout the field. It simply shows at a certain time we have higher SWC differences in deeper layers at different locations as compared to the upper layers.*

*In order to see if your suggestion could improve the calibration and ET estimation results, we calibrated the model from the deeper layers to the upper layers at TP 1 location. As you can see in the tables and figures below, during both calibration and validation periods the ET estimation had poorer results as shown by the RMSE.*

| Optimization from top layers to the lower layers | | | | | | | | | |
|---|---|---|---|---|---|---|---|---|---|
| Location | Depth (cm) | Calibration Period (2008-2010) | | | | Validation Period (2011-2012) | | | |
| | | R^2 | MAE | RMSE | NSE | R^2 | MAE | RMSE | NSE |
| Mead 1 | 10 | 0.542 | 0.024 | 0.036 | 0.533 | 0.532 | 0.016 | 0.033 | 0.503 |
| | 25 | 0.742 | 0.014 | 0.022 | 0.739 | 0.716 | 0.029 | 0.040 | 0.486 |
| | 50 | 0.409 | 0.013 | 0.023 | 0.407 | 0.603 | 0.041 | 0.074 | 0.157 |
| | 100 | 0.352 | 0.015 | 0.022 | 0.343 | 0.419 | 0.027 | 0.038 | 0.358 |

| Optimization from lower layers to the top layers | | | | | | | | | |
|---|---|---|---|---|---|---|---|---|---|
| Location | Depth (cm) | Calibration Period (2008-2010) | | | | Validation Period (2011-2012) | | | |
| | | R^2 | MAE | RMSE | NSE | R^2 | MAE | RMSE | NSE |
| Mead 1 | 10 | 0.466 | 0.024 | 0.039 | 0.438 | 0.591 | 0.018 | 0.034 | 0.471 |
| | 25 | 0.529 | 0.018 | 0.035 | 0.327 | 0.724 | 0.033 | 0.046 | 0.329 |
| | 50 | 0.405 | 0.017 | 0.028 | 0.121 | 0.677 | 0.037 | 0.060 | 0.456 |
| | 100 | 0.117 | 0.018 | 0.033 | -0.451 | 0.362 | 0.027 | 0.042 | 0.193 |

[Figure]

**Soil Hydraulic Parameters Optimization from top layers to the lower layers**

[Figure]

**Soil Hydraulic Parameters Optimization from lower layers to the top layers**

[revised manuscript text omitted]